# Influence Guided Context Selection for Effective Retrieval-Augmented Generation

**Jiale Deng, Yanyan Shen**[*]**, Ziyuan Pei, Youmin Chen, Linpeng Huang**
Shanghai Jiao Tong University
{jialedeng, shenyy, pzy_live, chenyoumin, lphuang}@sjtu.edu.cn

## Abstract

Retrieval-Augmented Generation (RAG) addresses large language model (LLM) hallucinations by grounding responses in external knowledge, but its effectiveness is compromised by poor-quality retrieved contexts containing irrelevant or noisy information. While existing approaches attempt to improve performance through context selection based on predefined context quality assessment metrics, they show limited gains over standard RAG. We attribute this limitation to their failure in holistically utilizing available information (query, context list, and generator) for comprehensive quality assessment. Inspired by recent advances in data selection, we reconceptualize context quality assessment as an inference-time data valuation problem and introduce the Contextual Influence Value (CI value). This novel metric quantifies context quality by measuring the performance degradation when removing each context from the list, effectively integrating query-aware relevance, list-aware uniqueness, and generator-aware alignment. Moreover, CI value eliminates complex selection hyperparameter tuning by simply retaining contexts with positive CI values. To address practical challenges of label dependency and computational overhead, we develop a parameterized surrogate model for CI value prediction during inference. The model employs a hierarchical architecture that captures both local query-context relevance and global inter-context interactions, trained through oracle CI value supervision and end-to-end generator feedback. Extensive experiments across 8 NLP tasks and multiple LLMs demonstrate that our context selection method significantly outperforms state-of-the-art baselines, effectively filtering poor-quality contexts while preserving critical information. Code is available at `https://github.com/SJTU-DMTai/RAG-CSM`.

## 1 Introduction

Retrieval-Augmented Generation (RAG) has emerged as a powerful approach for mitigating hallucinations in large language models (LLMs) by grounding their responses in external knowledge sources [4, 19, 24]. A typical RAG pipeline consists of two core components: a *retriever* that searches for query-relevant contexts from external knowledge sources, and an LLM *generator* that produces responses using the retrieved contexts. Despite its advantages, RAG faces important challenges in practical applications. That is, external knowledge sources may contain substantial noisy data, and retrievers based on similarity metrics are inherently imperfect [54]. As a result, retrieved contexts often include irrelevant and noisy information [12, 54, 57]. This issue is particularly problematic as LLM generators tend to rely heavily on the provided contexts [12, 31, 38], potentially producing incorrect responses when provided with poor-quality contexts.

Recent work [8, 20, 47, 57, 58] proposed context selection to shield generators from poor-quality contexts. This strategy depends on **context quality assessment**, which assigns a quality score to

---

[*]corresponding author.

39th Conference on Neural Information Processing Systems (NeurIPS 2025).

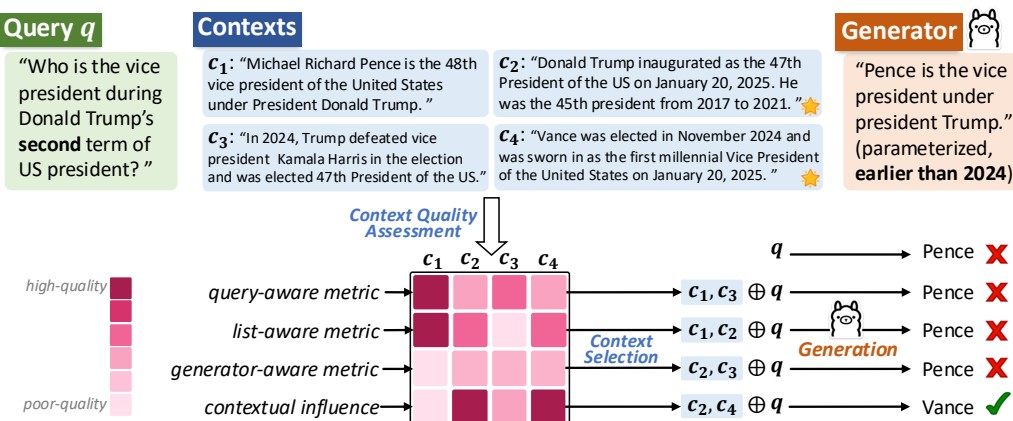

Figure 1: An example demonstrating different context quality metrics in practice. $c_2$ and $c_4$ are golden contexts containing crucial information about Trump's second presidency and his vice president Vance. Query-aware metrics favor $c_1$ and $c_3$ due to their mentions of "Trump" and "vice president". List-aware metrics score $c_2$ and $c_4$ higher by considering context relationships, but still favor $c_1$. Generator-aware metrics assign low scores to $c_1$ as it's redundant with LLM knowledge. CI value, by integrating all three dimensions, correctly identifies $c_2$ and $c_4$ as the most informative contexts.

each context to guide the selection process. Without loss of generality, the contexts are evaluated across three complementary dimensions: (1) **query-aware metric** that measures the semantic relevance between context and query, implemented through point-wise rerankers [8, 20, 37, 41]; (2) **list-aware metric** that considers relationships among multiple contexts, optimized through pair-wise and list-wise rerankers [29, 30, 34, 36, 53] to prompt diversity and complementarity in the selected contexts; and (3) **generator-aware metric** that evaluates how contexts align with the generator's existing knowledge, using metrics such as log likelihood [51] or mutual information with generated responses [47, 57]. However, our empirical studies in Section 6.1 demonstrate that these metrics achieve limited effectiveness in context selection, sometimes even reducing RAG performance [19]. As illustrated in Figure 1, query-aware and list-aware metrics lack generator feedback, potentially selecting contexts that duplicate or contradict generator's knowledge [12, 54]. Meanwhile, generator-aware metrics ignore inter-context relationships, risking the omission of critical information. Moreover, selection parameters such as top-$k$ (number of contexts to keep) must be specified in advance [18, 35], with optimal configurations varying significantly across different tasks, making it challenging to achieve ideal selection performance in practice.

Recent progress in training data valuation has shown promising results for selecting high-quality training samples and enhancing ML model performance [17, 44]. One key data valuation metric is data influence [22, 56], which quantifies a sample's importance by measuring the validation performance decrease when removed from the training set. Inspired by this, we reconceptualize context quality assessment as an **inference-time data valuation** problem and introduce the **Contextual Influence value (CI value)** for RAG context selection. Given a query $q$, context $c_i \in C$, and generator $f$, CI value is defined as $\phi_i(v) = v(f(q \oplus C)) - v(f(q \oplus \{C \backslash c_i\}))$, where $\oplus$ represents concatenation and $v(f(\cdot))$ is a utility function that measures generator output quality (e.g., EM or F1 scores). CI value naturally satisfies four key desiderata: (1) *query-awareness*: query-irrelevant contexts leads to $\phi_i(v) = 0$, indicating that CI value implicitly captures query-context relevance; (2) *list-awareness*: by measuring list-wise marginal contribution, CI value rewards unique and essential information while penalizing redundant content; (3) *generator-awareness*: with generator feedback, CI value effectively distinguishes between contexts that enhance generator performance and those that diminish it; (4) *ease-of-configuration*: instead of requiring task-specific top-$k$ tuning, CI value enables a straightforward selection strategy by keeping contexts with $\phi_i(v) > 0$, i.e., those whose removal degrades performance.

However, compared to data influence metrics in training data valuation, CI value computation faces two unique challenges. First, the utility evaluation $v(\cdot)$ depends on access to test labels, which are unavailable during inference. While some approaches attempt to estimate utility using model

confidence, such heuristics often prove unreliable in practice [7]. Second, computing exact CI values requires $n$ LLM forward passes for a $n$-length context list, substantially increasing inference latency. To address these challenges, we propose a CI Surrogate Model (CSM) that predicts CI values during inference. The CSM model is trained on the RAG training set and it can rapidly assign quality scores to contexts without requiring labels or multiple LLM calls. The approximation effectiveness of CSM depends on both its architecture and training strategy. Specifically, we employ a hierarchical structure that captures both local query-context relevance and global inter-context dependencies. For generator awareness, we explore two training strategies: (1) supervised learning using oracle CI values as targets, which provides implicit generator feedback; and (2) end-to-end training with the generator in the loop, which offers explicit signals about each context's impact.

We validate our framework through comprehensive experiments on 8 real-world NL tasks with 2 LLM backbones. Results demonstrate that our CI value surpasses existing context quality metrics in identifying high-quality contexts and streamlining selection configuration. Moreover, our proposed CSM achieves 15.03% average improvement in RAG generation performance over leading baselines.

## 2   Related Work

**Noise Robustness for RAG.** RAG systems often encounter poor-quality retrieval results containing irrelevant and noisy information [47, 57]. These poor-quality contexts not only distract LLMs [54] but can also lead to incorrect responses, as LLMs tend to overly trust external information [12] and struggle with the "lost-in-the-middle" problem [27] when processing lengthy contexts. To address these challenges, recent research has pursued two main approaches. The first approach enhances model capabilities through supervised fine-tuning [12, 47, 54] or instruction tuning [48, 55] to improve LLM noise robustness, or implements sophisticated pipelines like self-ask mechanisms [2] to guide LLM self-reflection. However, these solutions face practical limitations: fine-tuning LLMs is computationally expensive, and complex pipelines increase inference latency. The second approach employs external filters to rerank [30, 34, 50] or refine [8, 20, 47, 51, 57, 58] retrieval results, shielding generators from poor-quality contexts. These methods utilize LMs or LLMs as context selection models, training them through supervised [51] or reinforcement learning [57] based on quality metrics derived from prior knowledge, such as query relevance [8, 20, 50], log likelihood [51], mutual information [47, 57], and so on. However, as discussed in Section 1, these quality metrics lacks comprehensive utilization of available information (query, context list, and generator), leading to suboptimal selection performance.

**Inference-Time Data Valuation.** Data valuation metrics quantify each training example's contribution to model performance (e.g., its effect on validation accuracy), which is proven to be effective for data selection tasks that identify high-quality training samples to improve model performance. A fundamental approach is Leave-One-Out (LOO), which measures the performance degradation when removing a training sample, though it requires expensive retraining. Recent work has improved LOO through two main strategies: (1) influence-based methods [6, 14, 22, 56] that approximate LOO by using gradient and Hessian matrix without full retraining; (2) Shapley value-based methods [13, 17, 44] that enhance fairness by modeling complex sample interactions through cooperative game theory. Recently, inference-time data valuation has emerged as new direction of data valuation, focusing on assessing data quality of inference data [7]. However, one cannot directly compute the utility due to the unknown labels during inference. While simple heuristics like model confidence are proven to be unreliable [7], current research trains utility prediction models (UPMs) with regression objectives to estimate oracle utility. UPMs have been proven effective in data selection tasks of various domains [7, 33, 46, 56]. However, UPMs do not directly optimize the predicted data valuation scores against ground truth values, suffering the risk of error accumulation and inaccurate approximation.

## 3   Preliminaries

**Setup for RAG.** A typical RAG system consists of a retriever and an LLM generator $f$. Given a query $q$, the retriever retrieves a list of query-relevant contexts $C = \{c_1, ..., c_n\}$ from an external knowledge base. The generator then takes both the query and the retrieved contexts as input to generate an answer $\hat{y}$. Formally, this can be expressed as $\hat{y} = f(q \oplus C)$, where $\oplus$ denotes the combination of the query and the retrieved contexts, typically implemented as a simple concatenation.

**Utility Function.** We quantify the effectiveness of retrieved contexts using a utility function $v$: $2^n \rightarrow \mathbb{R}$, which maps any subset $S \subset C$ to a real-valued score reflecting its usefulness for answering the query. For NLP tasks, the utility function is typically defined by comparing the model's generated output against the ground-truth answers. Concretely, we set:

$$v_{f,q}(S) = \max_{y \in Y} -\mathcal{L}(y, f(q \oplus S)), \tag{1}$$

where $Y$ is the set of correct answers, and $\mathcal{L}$ is the cross entropy loss.

**Definition 1 (Contextual Influence)** *Given query $q$, retrieved context list $C$ and utility function $v$[2], the contextual influence value (CI value) for a context $c_i \in C$ is defined as:*

$$\phi_i(v) = v(C) - v(C \backslash c_i). \tag{2}$$

CI value quantifies the utility change when $c_i$ is removed from the context list. Unlike previous metrics, it simultaneously captures three key aspects: query-awareness (through $q$), list-awareness (through interactions within $C$), and generator-awareness (through feedback from $f$). A positive CI value ($\phi_i(v) > 0$) indicates that removing $c_i$ degrades utility (or increases test loss), suggesting that $c_i$ positively contributes to generation quality, and vice versa. To ensure fair CI value, we remove semantically duplicate contexts from $C$ before computing CI values.

**Context Selection for RAG.** It is commonly formulated as an optimization problem, whose objective is to maximize the utility of the generator based on the choice of retrieved contexts. Specifically, given $v$, the objective of context selection is to identify a subset $S_v^* \subset C$ that optimizes:

$$S_v^* = \arg\max_{S \subset C} v(S). \tag{3}$$

However, solving Equation (3) presents significant challenges. The utility function $v$ for the complex LLM generator lacks a tractable closed-form expression for analytical optimization. A brute-force approach that simply evaluates the utility of all possible subsets $v(S)$ would necessitate $2^n$ LLM generator forwards, which is computationally infeasible in practice as $n$ grows large.

**Context Selection via CI value.** To avoid the computationally intensive task of enumerating all possible subsets, we adopt a more practical approach by decomposing group influence into pointwise influence [11, 56]. Following previous research [44, 56], we aggregate contextual influences through summation: $\phi(v)[S] = \sum_{c_i \in S} \phi_i(v)$. Therefore, the context selection strategy based on CI values aims to maximize $\phi(v)[S]$ as a proxy for optimizing $v(S)$:

$$\hat{S}_{\phi(v)} = \arg\max_{S \subset C} \phi(v)[S]. \tag{4}$$

Since $\phi(v)[S] = \sum_{c_i \in S} \phi_i(v)$, $\hat{S}_{\phi(v)}$ consists of contexts with all positive CI values. That is, *when using CI value for context selection, we select all contexts with positive CI values.*

## 4 Methodology

### 4.1 CI Parameterization via Surrogate Model (CSM)

As established, directly computing CI values during inference is infeasible due to label dependency and computational cost, requiring a CI surrogate model (CSM) for approximation. Due to the query-awareness and list-awareness of CI value, our CSM should effectively capture both local query-context relevance and global list-level interactions. Inspired by recent advances in list-wise neural rerankers [26, 32], we design CSM with a hierarchical structure (Figure 2) comprising three components: (1) a local layer based on BERT-uncased [10] for query-context pair modeling; (2) a global layer with self-attention; (3) an MLP-based output layer. Given $q$ and $C = \{c_1, ..., c_n\}$, CSM first processes each query-context pair $(q, c_i)$ through the local layer to generate local embeddings $L = \{l_1, ..., l_n\}$, capturing semantic relationships between query and each context. These embeddings are then mean-pooled and fed into the global layer, where multi-head self-attention computes cross-context interactions to produce global embeddings $G = \{g_1, ..., g_n\}$. Finally, the output layer maps these global embeddings into relevance scores $M = \{m_1, ..., m_n\}$.

---

[2]We omit the subscripts $f$ and $q$ for simplicity.

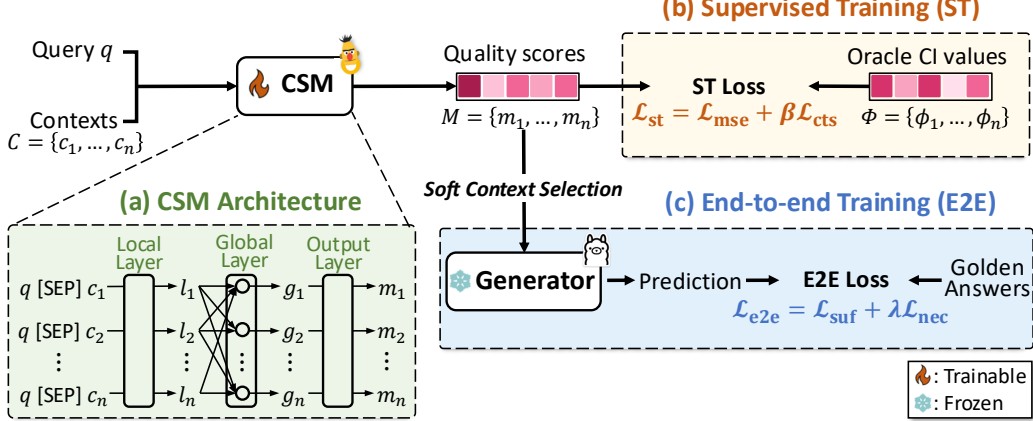

Figure 2: Overview of our proposed CSM: (a) CSM model architecture, (b) supervised training paradigm and (c) end-to-end training paradigm.

To effectively approximate CI value using CSM, we then introduce two training paradigms to establish generator awareness of CSM: (1) supervised training that implicitly encodes generator feedback through oracle CI values; (2) end-to-end training that directly propagates generator gradients through differentiable soft context selection.

## 4.2 Supervised Training of CSM

Supervised training establishes intrinsically generator awareness through CI value supervision. We create a training dataset $\mathcal{D} = \{d_1, ..., d_{|\mathcal{D}|}\}$ by collecting oracle CI values for all query-context pairs in RAG training dataset, where each sample $d_i = (q_i, Y_i, C_i, \Phi_i)$ contains a query $q_i$, the answer set $Y_i$, contexts $C_i = \{c_{i_1}, ..., c_{i_n}\}$, their oracle CI values $\Phi_i = \{\phi_{i_1}(v), ..., \phi_{i_n}(v)\}$. Following data valuation research [7, 46, 56], we frame CSM training as a supervised regression task. However, as shown in Apendix A, the CI value distribution is severely imbalanced, with approximately 80% of contexts having near-zero CI values, while samples with high-CI and low-CI contexts are very rare (16%). This significant imbalance makes CSM training particularly challenging, which we address through both data and loss perspectives.

**From the perspective of data**, we employ a combination of down sampling and data interventions. We first define the rarity rate of sample $d_i$ as $r_i := \mu(\Phi_i) + \alpha \cdot \sigma(\Phi_i)$, where $\mu(\cdot)$ and $\sigma(\cdot)$ represent the mean and standard deviation, respectively, and $\alpha$ is a balancing coefficient. Using two thresholds $\delta_1 < \delta_2$, we categorize samples into two distinct groups: (1) Trivial samples $\mathcal{D}_t$ ($r_{t_i} < \delta_1$ for $d_{t_i} \in \mathcal{D}_t$), which predominantly contain non-informative contexts; (2) Hard samples $\mathcal{D}_h$ ($r_{h_i} > \delta_2$ for $d_{h_i} \in \mathcal{D}_h$), which contain contexts with high-CI or low-CI contexts. For the majority of trivial samples, we apply down sampling to reduce their dominance in the dataset. For hard samples, we implement cross-instance intervention to balance the data distribution by increasing the number of samples with both high-CI and low-CI contexts. Due to space limitations, we focus on describing the intervention for constructing samples with high-CI contexts; the analogous process for samples with low-CI contexts can be found in Appendix B. The intervention process begins by collecting hard samples with high-CI contexts, denoted as $d_{h_i} = (q_i, C_i = \{C_i^P, C_i^N\}, \Phi_i)$, where $C_i^P = \{c_{i_k} | \phi_{i_k}(v) > \gamma\}$ represents high-CI contexts ($\gamma > 0$) and $C_i^N = C_i \setminus C_i^P$. We then sample another instance $d_j = (q_j, C_j, \Phi_j)$ whose query $q_j$ is semantically distinct from $q_i$. Following the rationale-environment recombination approach [49], we create a new sample by:

$$\hat{d}_{h_i} = (q_i, Y_i, \{C_i^P \cup \hat{C}_j\}, \hat{\Phi}_i), \tag{5}$$

where $\hat{C}_j$ is sampled from $C_j$. This intervention strategy is based on the intuition that when informative contexts ($C_i^P$) are placed in noisier environments (composed of $\hat{C}_j$ that are irrelevant to $q_i$), their marginal contribution to the context list becomes more pronounced [40], resulting in elevated CI values for $C_i^P$. Through this process, we effectively construct additional samples with high-CI contexts, thereby enriching the training set with more informative samples.

**From the perspective of training**, we implement a dual-loss strategy combining reweighted regression and contrastive learning. First, we mitigate majority class dominance through importance weighting:

$$\mathcal{L}_{\text{mse}} = \mathbb{E}_{d_i \in \mathcal{D}}[(S_i - \Phi_i)^2 / p(i)], \tag{6}$$

where $p(i)$ represents the empirical frequency of $r_i$ in the training distribution. For hard samples, we employ a contrastive loss term to enhance their discriminative signal:

$$\mathcal{L}_{\text{cts}} = -\mathbb{E}_{(q,Y,C,\Phi) \in \mathcal{D}_h} \mathbb{E}_{c \in C} \left[ \log \frac{\exp(g_c \cdot g_{c^+}/\tau)}{\sum_{c^- \in C^-} \exp(g_c \cdot g_{c^-}/\tau) + \exp(g_c \cdot g_{c^+}/\tau)} \right], \tag{7}$$

where anchor $c$ is a context from hard samples, positive context $c^+$ shares similar CI value with $c$ ($|\phi_c(v) - \phi_{c^+}(v)| < \epsilon_1$), negative contexts $C^-$ have divergent CI values from $c$ ($|\phi_c(v) - \phi_{c^-}(v)| > \epsilon_2$), and $\tau$ is a temperature hyperparameter. The final supervised training loss is the linear combination controlled by a hyperparameter $\beta$: $\mathcal{L}_{\text{st}} = \mathcal{L}_{\text{mse}} + \beta \mathcal{L}_{\text{cts}}$.

### 4.3 End-to-end Training of CSM

End-to-end training explicitly injects generator awareness by directly using the generator's output as a signal to optimize CSM's parameters. We denote the training set of end-to-end training by $\mathcal{E} = \{(q_i, Y_i, C_i)\}$. A typical end-to-end training paradigm requires computing values for each context, selecting contexts $S = \{c_i | \phi_i(v) > 0\}$ based on these values, and feeding the selected contexts to the generator for final answer generation. However, this context selection process is discrete and non-differentiable. To address this, we treat CSM's output $M$ as a mask and implement **soft context selection** during training by masking the generator's input tokens with $M$. We employ the Gumbel-Softmax trick [9, 28, 40] to approximate a binary mask, i.e., $\hat{M} = \text{Gumbel}(M)$. The generator's masked input is then reconstructed as: $H = H_q \oplus H_c$, where $H_q = f_{\text{tok}}(q)$ represents the tokenized query and $H_c = \hat{M} \odot f_{\text{tok}}(C)$ denotes the masked tokenized context, with $f_{\text{tok}}(\cdot)$ being the generator $f$'s tokenizer. Additionally, we construct the complementary masked tokenized context as $H_t = (1 - \hat{M}) \odot f_{\text{tok}}(C)$, which effectively removes high-value contexts before tokenization. Then, to align mask values with CI value, we design following loss terms:

$$\mathcal{L}_{\text{suf}} = -\mathbb{E}_{(q,Y,C) \in \mathcal{E}}[Y^T \log(f(H_q \oplus H_c)], \tag{8}$$

$$\mathcal{L}_{\text{nec}} = \mathbb{E}_{(q,Y,C) \in \mathcal{E}}[\text{KL}(Y_{\text{unif}}, f(H_q \oplus H_t)], \tag{9}$$

where $\text{KL}(\cdot)$ denotes the KL-Divergence and $Y_{\text{unif}}$ represents the uniform distribution. $\mathcal{L}_{\text{suf}}$ encourages the generator to produce accurate responses when high-value contexts are provided, ensuring that these selected contexts contain sufficient information for answer correctly. Meanwhile, $\mathcal{L}_{\text{nec}}$ penalizes the CSM if the generator still generates correct answers even after the removal of high-value contexts, thereby discouraging false positive selections in the context selection process and ensuring that the high-value contexts are necessary for generating correct answers. The overall end-to-end training loss effectively approximates the CI value by combining both sufficiency and necessity through a linear combination controlled by hyperparameter $\lambda$: $\mathcal{L}_{\text{e2e}} = \mathcal{L}_{\text{suf}} + \lambda \mathcal{L}_{\text{nec}}$.

## 5 Experimental Setup

**Tasks and Datasets.** We consider the following knowledge-intensive NLP tasks: (1) Open-Domain QA, including NQ [23], TriviaQA [21] and WebQA [3]. (2) Multihop QA that requires multi-step reasoning to generate answers, including HotpotQA [52] and 2WikiMultiHopQA [15]. (3) Fact Checking dataset FEVER [43] that challenges the model to use complex reasoning to determine the factual accuracy of given claims. (4) Multiple Choice dataset TruthfulQA [25]. (5) Long-Form QA dataset ASQA [39] that generating long and abstract answers given the question. Following [19, 47], we report Exact Match (EM) for Open-Domain QA datasets, F1 for Multihop QA and Long-Form QA datasets, and Accuracy for Fact Checking and Multiple Choice datasets.

**Baselines.** We consider the following baselines: (1) **Vanilla LLM**: LLM without retrieval augmentation. (2) **Standard RAG**: a sequential RAG pipeline using FlashRAG [19] with retrieval (preserving all retrieved contexts). (3) **bge-reranker** [50]: a context selection baseline based on query-aware quality metrics. It leverages a cross-encoder to perform point-wise context reranking. (4) **RankGPT** [41]: a context selection baseline based on list-aware quality. It is a list-wise reranker

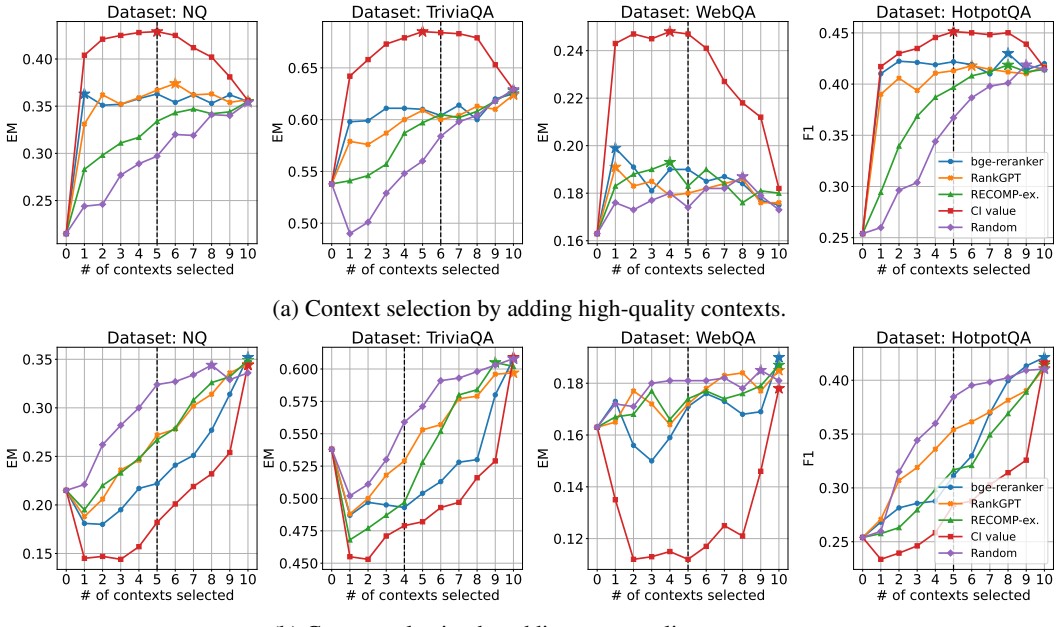

(a) Context selection by adding high-quality contexts.

(b) Context selection by adding poor-quality contexts.

Figure 3: RAG generation performance when selecting contexts using different quality metrics. (a) Selecting high-quality contexts, where higher curve indicates better metric. (b) Selecting poor-quality contexts, where lower curve indicates better metric. Dashed line marks the top-$k$ cutoff where the average CI value is zero and star marks the top-$k$ yielding best performance. For baselines, we use predicted scores for bge-reranker and RankGPT, and oracle log likelihood scores for RECOMP-ex.

that feeds the whole list into LLM and generates a ordered context list based on their relevance to query. We use the generator LLM as the lit-wise reranker in our experiments. (5) **RECOMP-ex** [51]: a context selection baseline based on generator-aware quality metrics. Given query $q$, context $c_i$ and ground truth answer $y$, the quality score of $c_i$ is $\log p_f(y|[q \oplus c_i])$. It employs the oracle scores train a BERT-based context selection model with contrastive learning. (6) **RECOMP-abs** [51]: a context summarization baseline by distilling a lightweight abstractive compressor from extreme-scale teacher LLMs like GPT-3.5. (7) **Ret-Robust** [54]: a generator enhancing baseline by fine-tuning the LLM via LoRA [16] to be robust to external noise. (8) **Self-RAG** [2]: an agentic baseline that performs on-demand retrieval and learns to reflect on retrieved contexts while critiquing generated answers. (9) **RQ-RAG** [5]: an agentic baseline that enhances the RAG pipeline through explicit rewriting, decomposition, and disambiguation. (10) **oracle CI value**: context selection based on oracle CI value, and (11) **CSM-st** and **CSM-e2e**: the CI Surrogate Model for context selection, trained by $\mathcal{L}_{\mathrm{st}}$ and $\mathcal{L}_{\mathrm{e2e}}$, respectively.

**Implementation Details.** To demonstrate the versatility of our method, we choose two backbones differing in architecture: Llama3-8b-intruct [1] and Qwen2.5-7b-instruct [42]. For the retrieval corpus, we utilize the Wikipedia dump from December 2018, and pre-process it into chunks (100 words per chunk). We use E5-base-v2 [45] as the dense retriever and retrieve the top 10 chunks from all Wikipedia chunks. We follow the FlashRAG benchmark[19] for data, splits and baselines (including Vanilla LLM, Standard RAG, bge-reranker and RECOMP). We conducted all the experiments on a server equipped with Montage Jintide(R) C6226R CPU, 256GB Memory, and 4 Nvidia GeForce RTX 4090 GPUs. Detailed setup of our method can be found in Appendix C.3.

# 6 Experimental Results

## 6.1 Effectiveness of CI value

This part of experiments aims to prove that CI value serves as an effective metric for context selection, as it eliminates the need for complex top-$k$ configuration and directly correlates with RAG generation

Table 1: RAG generation performance (%) on 8 downstream tasks with different baselines. The best results are in **bold** and the second best are with underscore. The oracle CI values with asterisk superscript act as a performance reference of our proposed CSM.

| Task type | NQ | TriviaQA | WebQA | HotpotQA | 2Wiki | FEVER | TruthfulQA | ASQA |
| --- | --- | --- | --- | --- | --- | --- | --- | --- |
| | | Open-Domain QA | | Multihop QA | | Fact Check. | Multiple Choice | Long-Form QA |
| | | (EM) | | (F1) | | (Acc) | (Acc) | (F1) |
| **Llama3-8B** | | | | | | | | |
| Vanilla LLM | 20.58 | 52.87 | 16.39 | 24.27 | 23.51 | 71.52 | 30.11 | 31.34 |
| Standard RAG | 37.01 | 62.36 | 18.21 | 40.95 | 24.38 | 90.76 | 27.05 | 34.70 |
| bge-reranker | 39.06 | 64.17 | 18.85 | 41.96 | 25.92 | 90.57 | 28.64 | 33.97 |
| RankGPT | 38.61 | 61.83 | 19.24 | 41.53 | **27.26** | 78.58 | 30.11 | **35.21** |
| RECOMP-ex | 29.86 | 60.67 | 18.36 | 39.06 | 24.55 | - | - | - |
| RECOMP-abs | 32.85 | 58.77 | 18.70 | 39.94 | 25.57 | 90.66 | 30.60 | 34.02 |
| Ret-robust | 41.77 | 65.83 | 19.76 | 45.69 | 25.91 | 90.69 | 27.05 | 34.50 |
| Self-RAG | 36.23 | 38.26 | 21.83 | 29.98 | 25.43 | 85.77 | 29.75 | 32.56 |
| RQ-RAG | 34.27 | 55.31 | **26.12** | 35.22 | 26.08 | 90.13 | 27.36 | 33.64 |
| CSM-st | **42.53** | **69.59** | 24.77 | **47.53** | 25.97 | **91.39** | 30.97 | 34.75 |
| CSM-e2e | 41.61 | 67.88 | 26.05 | 45.61 | 26.48 | 90.74 | 30.56 | 33.68 |
| oracle CI val. | 45.79* | 71.98* | 27.81* | 48.28* | 30.72* | 94.58* | 32.09* | 35.70* |
| **Qwen2.5-7B** | | | | | | | | |
| Vanilla LLM | 14.88 | 41.75 | 16.54 | 26.48 | 29.44 | 79.61 | 27.01 | 30.21 |
| Standard RAG | 38.50 | 63.29 | 21.51 | 44.81 | 33.68 | 91.14 | 23.26 | 33.46 |
| bge-reranker | 39.53 | 63.76 | 21.70 | 45.59 | 34.48 | 91.05 | 25.34 | 33.00 |
| RankGPT | 39.14 | 63.07 | 21.21 | 45.70 | 35.81 | 90.20 | 25.83 | 34.98 |
| RECOMP-ex | 35.76 | 61.08 | 20.28 | 42.04 | 32.47 | - | - | - |
| RECOMP-abs | 31.80 | 59.26 | 20.37 | 42.27 | 33.57 | 91.33 | **32.07** | 34.70 |
| Ret-robust | 42.77 | 64.65 | 28.52 | 44.20 | 36.98 | 91.10 | 23.25 | 33.57 |
| Self-RAG | 44.93 | 63.29 | 28.48 | 45.69 | 43.27 | 90.05 | 27.17 | 33.28 |
| RQ-RAG | 45.72 | 64.33 | 26.47 | 49.62 | 42.75 | 91.27 | 28.06 | 34.59 |
| CSM-st | **47.38** | 65.26 | **28.54** | **51.95** | **48.67** | **92.98** | 28.77 | 34.05 |
| CSM-e2e | 46.19 | **66.35** | 26.38 | 49.44 | 47.23 | 91.02 | 27.72 | **35.62** |
| oracle CI val. | 50.64* | 69.19* | 30.61* | 53.78* | 49.08* | 94.63* | 28.56* | 36.72* |

performance: selecting high-CI contexts improves the performance while low-CI contexts degrade it. Case studies and detailed experiments on other datasets are provided in Appendix D and E.

**Top-$k$ Configuration.** As illustrated in Figure 3a, leveraging CI value as context selection metric eliminates the need for top-$k$ tuning, as simply preserving contexts with positive CI values consistently achieves optimal or near-optimal RAG performance. In contrast, other metrics require dataset-specific top-$k$ configurations, with optimal values varying significantly across datasets (e.g., bge-reranker's optimal top-$k$ ranges from 1 for NQ to 10 for TriviaQA). This dataset-dependent variation makes it challenging to determine a universal top-$k$ value that performs well across different datasets, highlighting the practical advantage of CI value in context selection.

**Context Selection by Adding High-Quality Contexts.** The high-quality selection experiment is performed with the following steps [17]: For each context quality metric, we preserve a candidate context set $S$. $S = \varnothing$ initially. We select contexts from the retrieved contexts in descending order of the quality scores and add them to $S$. Each time the contexts are selected, we leverage current $S$ as the reference of generator to answer user queries, evaluate the answer quality using metrics in Section 5 and plot the performance curve. In an ideal scenario, selecting the most helpful contexts first should produce a sharp initial performance increase, followed by a decline as lower-quality contexts are included in $S$. Figure 3a illustrates the RAG performance curves from this experiment, with higher curves indicating superior quality metrics. All metrics outperform the random baseline, confirming their effectiveness in identifying high-quality contexts. CI value-based selection shows a consistent pattern aligning with the ideal pattern across datasets, while other baselines exhibit fluctuating performance as $S$ grows.

**Context Selection by Adding Poor-Quality Contexts.** We conduct poor-quality context selection experiments following a similar procedure to high-quality selection, but with contexts added in ascending order of their quality scores. In contrast to high-quality selection, the ideal performance curve should initially decline sharply as poor-quality contexts are incorporated into $S$, then gradually improve as higher-quality contexts are added. Figure 3b illustrates the RAG performance curves from this experiment, with lower curves indicating superior quality metrics. All metrics outperform the random baseline, confirming their effectiveness in identifying poor-quality contexts. Experimental

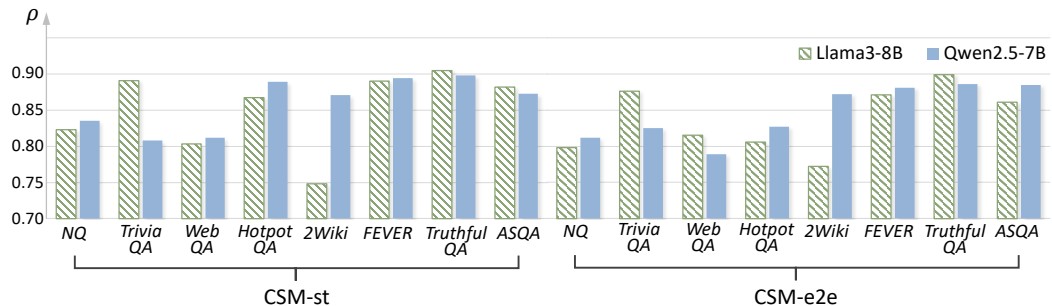

Figure 4: The Spearman correlation ($\rho$) of CSM's predictions with the oracle CI values.

results indicate that CI value's effectiveness in identifying poor-quality contexts, since its performance curves are consistently and significantly lower than other baselines.

## 6.2 Effectiveness of CI Surrogate Models

These experiments evaluate CSM's effectiveness in improving RAG performance through context selection, its ability to approximate oracle CI values, and the contributions of different CSM modules.

**Overall Generation Performance.** Table 1 presents a comprehensive comparison of RAG generation performance across different baselines. For fair comparison between context selection methods, we set top-$k = 5$. Our proposed CSM demonstrates significant improvements across nearly all tasks: both CSM-st and CSM-e2e achieve the best or second-best results on all eight tasks. In Open-Domain QA, CSM outperforms all baselines by 21.72% in EM, and it achieves a 19.40% F1 improvement in Multihop QA, highlighting the crucial role of high-quality contexts in generating correct answers. Although improvements on TruthfulQA and ASQA are more modest, CSM still ranks first or second on these tasks. The bge-reranker performs well on simple QA but struggles in complex scenarios (e.g., Multihop QA and Long-Form QA), while list-wise RankGPT shows better performance than bge-reranker in these challenging settings, emphasizing the importance of modeling context interactions. RECOMP-ex's performance is sometimes even inferior (e.g., 1.7% lower on ASQA), revealing the limitations of relying solely on generator feedback for context selection. Ret-Robust emerges as a strong baseline by enhancing generator at the cost of expensive LLM fine-tuning. It is worth noting that CSM outperforms advanced agentic baselines (e.g., Self-RAG, RQ-RAG) on most tasks. This demonstrates that filtering out low-quality contexts yields greater benefits compared to sophisticated agentic approaches that rely on complex reflection and planning mechanisms.

**Approximation Effectiveness.** We evaluate CSM's effectiveness in approximating oracle CI values using Spearman rank correlation, which measures the strength of monotonic relationships through the Pearson correlation of ranked values (ranging from -1 to 1, with 1 indicating perfect positive correlation). As shown in Figure 4, both CSM-st and CSM-e2e consistently achieve correlation coefficients above 0.75 across all tasks for both Llama3-8B and Qwen2.5-7B models, demonstrating strong alignment between CSM's predictions and oracle CI values.

**Ablation Studies.** We conduct ablation studies on both CSM-st and CSM-e2e variants to validate our training strategies, and present the results in Table 2. For CSM-st, removing data intervention causes an average performance drop of 11.98%↓, highlighting the importance of cross-instance intervention in creating balanced training samples. The removal of contrastive loss leads to an average 8.04%↓ decrease, demonstrating its effectiveness in enhancing supervision for hard samples. For CSM-

Table 2: Ablations on training strategy of CSM.

|  | NQ | TriviaQA | WebQA | HotpotQA |
|---|---|---|---|---|
| **CSM-st** | 42.53 | 69.59 | 24.77 | 47.53 |
| w/o interv. | 37.67 | 62.91 | 19.23 | 45.38 |
| w/o $\mathcal{L}_{cts}$ | 40.82 | 65.39 | 20.19 | 45.82 |
| **CSM-e2e** | 41.61 | 67.88 | 26.05 | 45.61 |
| w/o $\mathcal{L}_{suf}$ | 35.44 | 63.39 | 22.79 | 42.34 |
| w/o $\mathcal{L}_{nec}$ | 36.85 | 62.74 | 21.35 | 42.57 |

e2e, ablating either $\mathcal{L}$suf or $\mathcal{L}$nec results in significant performance drops (10.28%↓ and 10.93%↓, respectively), showing their complementary roles in guiding CSM to retain high-quality contexts while filtering out poor-quality ones.

**Inference-time Latency.** The efficiency of RAG systems primarily depends on the parameter size of the context selection model when keeping the retriever and generator the same. Table 3 presents CUDA times (from context selection to generation) of different context selection baselines w.r.t. the number of retrieved contexts. Since CSM selects all positive contexts, for fair comparison, we set the number of pre-

Table 3: Inference-time latency (ms) comparison between CSM and baselines.

|  | NQ | | TriviaQA | | HotpotQA | |
| --- | --- | --- | --- | --- | --- | --- |
|  | $n=10$ | $n=50$ | $n=10$ | $n=50$ | $n=10$ | $n=50$ |
| Standard RAG | 320 | 811 | 252 | 810 | 261 | 814 |
| RankGPT | 874 | 1437 | 779 | 1561 | 741 | 1640 |
| RECOMP-abs | 299 | 662 | 254 | 721 | **202** | 924 |
| CSM | **253** | **481** | **192** | **402** | 206 | **423** |

served contexts (i.e., top-$k$) for the baselines equal to the average number of contexts with positive CI values (i.e., $k_{pos}$). For $n = 10$, we set top-$k = k_{pos} = 5$, and for $n = 50$, we set top-$k = k_{pos} = 23$. Compared to baseline methods, CSM achieves significantly lower inference latency by leveraging the lightweight model architecture detailed in Figure 2. This efficiency advantage becomes increasingly pronounced as the number of retrieved contexts increases from $n = 10$ to $n = 50$. For example, on the NQ dataset with $n = 50$, CSM reduces latency to 481 ms, far outperforming RankGPT (1437 ms) and RECOMP-abs (662 ms) under the same condition. This result clearly demonstrates that CSM scales more efficiently with the growth of $n$, maintaining low latency even when handling a larger volume of retrieved contexts.

## 7   Conclusion

This paper introduces the Contextual Influence (CI) value, a novel metric for selecting high-quality contexts that enhance RAG performance. The CI value improves upon existing metrics by simultaneously possessing four desirable properties, i.e., query-awareness, list-awareness, generator-awareness and ease-of-configuration. We propose a parameterized surrogate model (CSM) to predict CI values during inference. To ensure high prediction accuracy, CSM features a hierarchical architecture that evaluates both query-context relevance and interactions between different contexts. We explore two approaches to optimizing CSM, i.e., supervised learning using oracle CI values and end-to-end training incorporating generator feedback. Empirical studies across 8 NLP tasks and 2 LLM backbones demonstrate that the CI value effectively distinguishes high-quality contexts from lower-quality ones, and our proposed CSM outperforms context selection baselines with an average RAG generation performance improvement of 15.03%. While CSM is effective and lightweight, its training remains challenging and currently requires task-specific optimization. Future research should focus on developing a universal context selector capable of generalizing across different tasks.

## 8   Acknowledgements

This work is supported by the National Key Research and Development Program of China (No. 2024YFB4505203), National Natural Science Foundation of China (No. 62522211, No. 62202255), and Key Research and Development Program of Xinjiang Uygur Autonomous Region (Grant No. 2023B01027, 2023B01027-1).

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

# A  Distribution Analysis for CI Value

We first analyze the distribution of CI values. For a given context, the larger the absolute value of its CI value, the greater its impact on RAG performance (both positive and negative impacts). Conversely, when the absolute value approaches zero, it indicates that the context has minimal influence on RAG performance. Figure 5 illustrates the CI value distribution across different datasets, where the x-axis represents the scaled CI value (we scale CI values into the range of $[-1, 1]$ without changing their relative ranking) and the y-axis shows the number of contexts. The CI value distribution exhibits severe imbalance, with the majority of contexts having near-zero CI values. Taking NQ as an example, 77.94% of contexts have absolute CI values lower than 0.1, indicating that a substantial portion of contexts contribute little to RAG generation performance. These contexts are likely query-irrelevant or redundant with the generator's parameter knowledge. In NQ, only 3.40% of contexts have absolute CI values higher than 0.3, suggesting that very few contexts significantly influence RAG performance, either positively or negatively. However, these contexts are precisely the key ones that we need to either select or eliminate to optimize RAG performance.

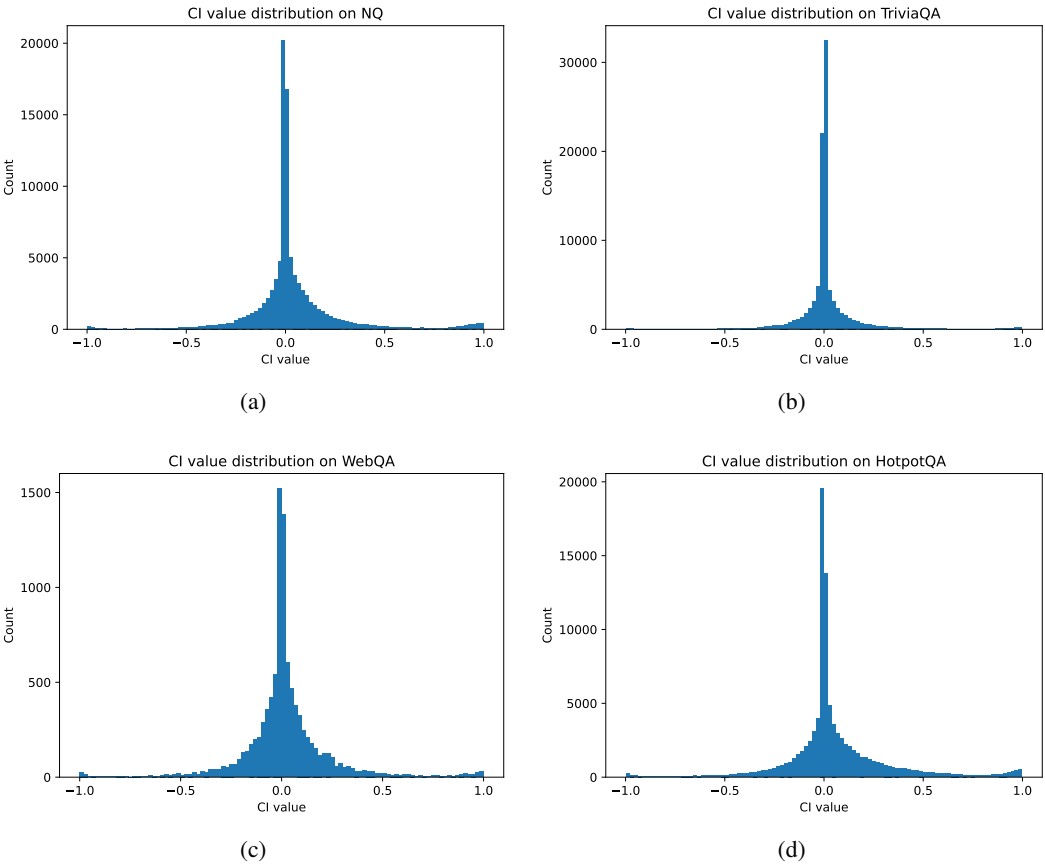

Figure 5: CI value distribution on different datasets, with Llama3-8B as generator.

We then analyze the distribution of rarity rates for supervised training samples in Figure 6. The rarity rate of a sample $d_i$ is defined as $r_i := \mu(\Phi_i) + \alpha \cdot \sigma(\Phi_i)$, where $\mu(\cdot)$ and $\sigma(\cdot)$ represent the mean and standard deviation, respectively, and $\alpha$ is a balancing coefficient set to 10. To categorize the training samples, we employ two thresholds $\delta_1$ and $\delta_2$ (where $\delta_1 < \delta_2$), dividing them into two distinct sets: the trivia sample set $\mathcal{D}_t$ and the hard sample set $\mathcal{D}_h$. Specifically, any sample $d_{t_i} \in \mathcal{D}_t$ satisfies $r_{t_i} < \delta_1$, while any sample $d_{h_i} \in \mathcal{D}_h$ satisfies $r_{h_i} > \delta_2$. In our experiments, we set $\delta_1 = \delta_2 = 5$. Hard samples are characterized by containing contexts with relatively high or low CI values within their corresponding context lists. However, the proportion of hard samples is notably small. Taking NQ as an example, only $|\mathcal{D}_h|/|\mathcal{D}| = 15.61\%$ of the samples fall into the hard sample category, indicating that a small fraction of samples contain contexts that are either highly beneficial or detrimental to RAG generation performance.

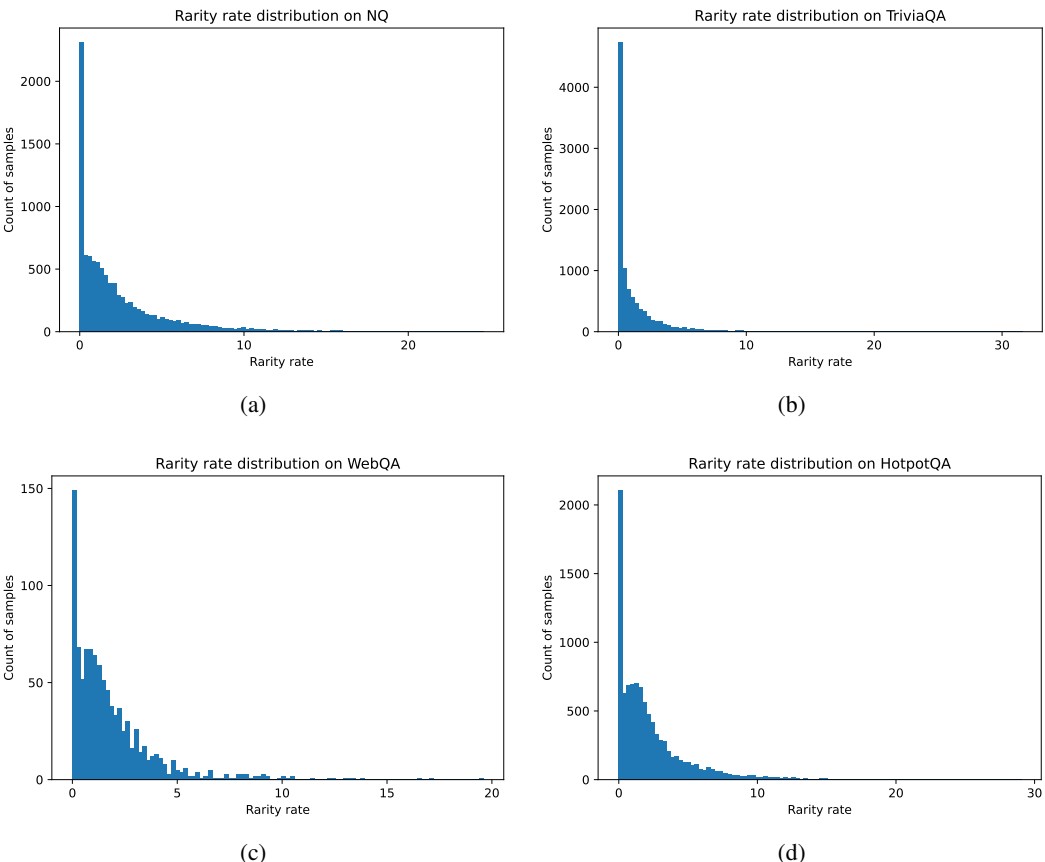

Figure 6: Rarity rate distribution on different datasets, with Llama3-8B as generator.

In summary, through our analysis of CI value distribution in real-world data, we observe that only a small fraction of contexts exhibit either high or low CI values, and these contexts are concentrated in a limited number of training samples. However, these underrepresented contexts are precisely the ones whose patterns we need to learn, as they represent the contexts that significantly influence RAG performance. The severe imbalance in the data distribution poses significant challenges for our CSM's generalization capabilities within the supervised learning paradigm. To tackle this issue, we introduce a novel solution that simultaneously addresses the imbalance problem from both data and loss perspective.

## B  Data Intervention

In this section, we present our approach for performing data intervention to increase the number of hard samples containing low-CI contexts. We begin by collecting hard samples with high-CI contexts, denoted as $d_{h_i} = (q_i, C_i = \{C_i^P, C_i^N\}, \Phi_i)$, where $C_i^P = \{c_{i_k} | \phi_{i_k}(v) > \gamma\}$ represents the set of high-CI contexts ($\gamma > 0$) and $C_i^N = C_i \backslash C_i^P$ represents the remaining contexts. Next, we sample another instance $d_j = (q_j, C_j, \Phi_j)$ whose query $q_j$ is semantically distinct from $q_i$. We then construct a new sample through the following intervention:

$$\hat{d}_{h_j} = (q_j, Y_j, \{C_i^P \cup \hat{C}_j\}, \hat{\Phi}_j), \tag{10}$$

where $\hat{C}_j$ is a subset sampled from $C_j$. Since contexts in $Ci^P$ are considered positively relevant to query $q_i$, and given that $q_j$ is semantically distinct from $q_i$, these contexts should be considered irrelevant for $q_j$ and their corresponding CI values should be low. This intervention process effectively generates additional samples containing low-CI contexts, thereby enriching our training set with more informative samples that better represent the challenging cases encountered in real-world scenarios.

Table 4: Detailed data statistics for CSM training.

| Task | Dataset Name | #train | #Dev | #Test |
|---|---|---|---|---|
| Open-Domain QA | NQ | 79,168 | 8,757 | 3,610 |
| | TriviaQA | 78,785 | 8,837 | 11,313 |
| | WebQA | 3,022 | 756 | 2,032 |
| Multihop QA | HotpotQA | 72,357 | 18,090 | 7,405 |
| | 2Wiki | 12,000 | 3,000 | 12,576 |
| Fact Checking | FEVER | 83,972 | 20,994 | 10,444 |
| Multiple Choice | TruthfulQA | 327 | 82 | 408 |
| Long-Form QA | ASQA | 3,482 | 871 | 948 |

Table 5: Comparison between different RAG context selection methods from their design principles.

| Method | Query-awareness | List-awareness | Generator-awareness | Ese-of-configuration |
|---|---|---|---|---|
| bge-reranker | ✓ | ✗ | ✗ | ✗ |
| RankGPT | ✓ | ✓ | ✗ | ✗ |
| RECOMP-ex | ✓ | ✗ | ✓ | ✗ |
| RECOMP-abs | ✓ | ✓ | ✗ | ✗ |
| CSM | ✓ | ✓ | ✓ | ✓ |

## C  Detailed Experimental Setup

### C.1  Dataset Details

Table 4 presents the dataset statistics, which are publicly available from [19]. For datasets without a provided test set, we utilize the development set as the test set and perform a split on the training set, allocating 80% as training set and 20% as dev set. Note that for experiments in Section 6.1 and Appendix D, we evaluate the context selection experiments (adding high/poor-quality contexts) on the first 1000 test samples. As demonstrated in [19], the baseline performance on this subset closely mirrors the performance on the complete test set.

### C.2  Baseline Setup

In Table 5, we present a detailed comparison of various context selection methods, emphasizing their real-world applicability. This comparison focuses on four key aspects: (1) Query-awareness: whether the method incorporates query-context relevance in measuring context quality. (2) List-awareness: whether the method considers the context list information in measuring context quality. (3) Generator-awareness: whether the method takes into account generator feedback in measuring context quality. (4) Ease-of-configuration: whether the method eliminates the need for tuning the hyperparameter top-$k$ across different tasks. These aspects collectively ensure the practical usability of the methods in real-world scenarios. We follow [19] for setting up baselines, whose details can be found in the official website[3].

### C.3  Implementation Details of CSM

**CSM Model Architecture**. Our model architecture consists of three main components: (1) a pre-trained BERT-uncased [10] model serving as the local layer, (2) a global layer comprising 3 layers of 8-head self-attention layers, and (3) a 2-layer MLP functioning as the output layer.

**RAG Pipeline**. We implement a sequential RAG pipeline following FlashRAG [19]. The retrieval source is the Wikipedia dump from December 2018, which we preprocess into chunks of 100 words each. For each query, we retrieve 10 chunks using a dense retriever based on the E5-base-v2 [45]

---

[3]`https://github.com/RUC-NLPIR/FlashRAG/blob/main/docs/original_docs/baseline_details.md`

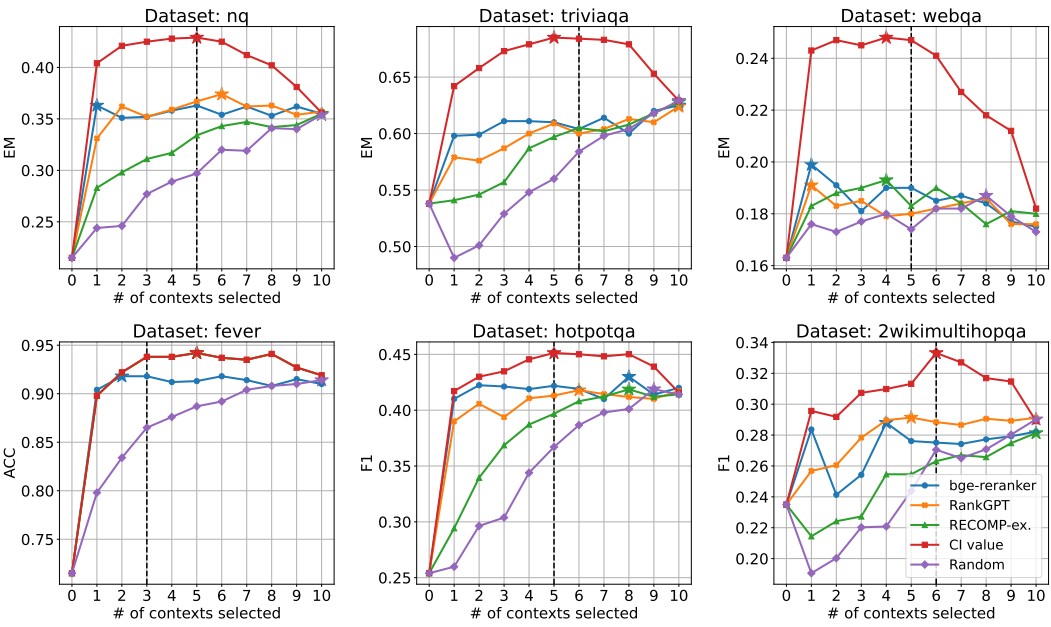

Figure 7: RAG generation performance when selecting high-quality contexts using different quality metrics, where higher curve indicates better metric. Dashed line marks the top-$k$ cutoff where the average CI value is zero and star marks the top-$k$ yielding best performance. For baselines, we use predicted scores for bge-reranker and RankGPT, and oracle log likelihood scores for RECOMP-ex.

model. We use Llama3-8b-intruct [1] and Qwen2.5-7b-instruct [42] as the LLM generators. All experiments are conducted with a fixed random seed of 2024 for reproducibility.

**Hyperparameter setting**. In our experiments, we employ the following hyperparameters: for supervised training, we set $\tau = 1$ and $\beta = 0.1$ and train CSM for 10 epochs with a batch size of 16; for end-to-end training, we set $\lambda = 1$ and train CSM for 10 epochs with a batch size of 4.

# D  Additional Experiments on Effectiveness of CI Value

In Figure 7 and Figure 8, we present additional experiments focusing on context selection by adding high-quality contexts and poor-quality contexts, respectively. Our comprehensive experiments across diverse datasets consistently show that employing CI value as a quality metric for context selection proves to be an effective strategy, successfully identifying crucial contexts while simultaneously eliminating detrimental ones.

# E  Case Studies

To illustrate the efficacy of the CI value metric for context selection in a RAG system, we present specific case studies in this section.

## E.1  Case Study 1

As illustrated in Figure 9, this case encompasses retrieved contexts for a query from NQ dataset and the corresponding predicted answers under different conditions. In this case, the CI value can identify valid information, enabling the generator to produce the correct answer. Our key observations are as follows: When all retrieved contexts ($c_1$, $c_2$, $c_3$, and $c_4$) are supplied to the LLM, contexts with negative CI values significantly distort the generated response. Specifically, $c_1$, characterized by a CI value of $\phi_1(v) = -0.19$, incorrectly conflates Toyota's arrival in El Salvador in 1953 with its entry into the United States market, that leads to incorrect prediction "May 1953". Similarly, $c_2$ with a CI value of $\phi_2(v) = -0.14$, misrepresents Toyota's initial entry into the U.S. market as an export event

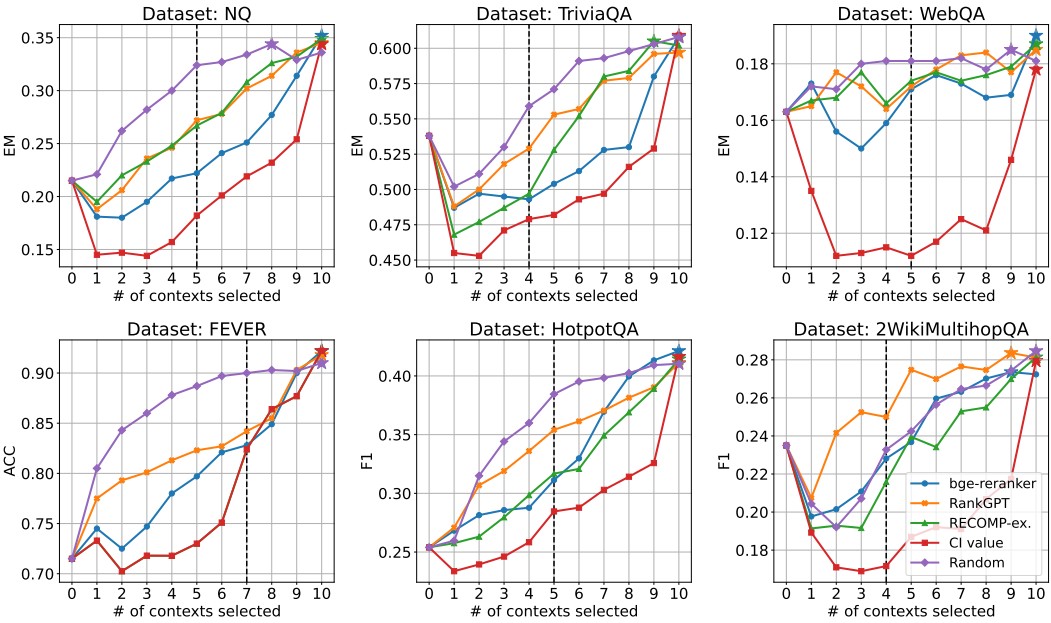

Figure 8: RAG generation performance when selecting poor-quality contexts using different quality metrics, where higher curve indicates better metric. Dashed line marks the top-$k$ cutoff where the average CI value is zero and star marks the top-$k$ yielding best performance. For baselines, we use predicted scores for bge-reranker and RankGPT, and oracle log likelihood scores for RECOMP-ex.

in June 1958. Meanwhile, $c_4$, with $\phi_4(v) = 0$, focuses on Toyota's early corporate history, which is tangential to the query regarding its U.S. market entry, thus rendering it contextually irrelevant. These query-irrelevant contexts occupy the context window of LLM and increase the inference time without improving its performance. In contrast, when only the context with a positive CI value, $c_3$ ($\phi_3(v) = 1.01$), which correctly states that Toyota entered the American market in 1957, is included in the input, the LLM produces the accurate answer. This case study underscores the critical role of CI values in filtering out deleterious and extraneous information, thereby enabling more accurate and reliable predictions in RAG frameworks.

### E.2 Case Study 2

Figure 10 presents another illustrative example from the NQ dataset. When all retrieved contexts are provided to the LLM, contexts with negative CI values significantly distort the generated response, leading to the incorrect answer "Cars". Our analysis reveals that context $c_2$ is the primary cause of the incorrect generation, as it contains mentions of "Cars" with the Oscar reward, resulting in a negative CI value of $\phi_2(v) = -0.14$. In contrast, context $c_1$, which contains the correct answer "Ratatouille", receives a relatively high CI value of $\phi_1(v) = 0.57$. Meanwhile, context $c_3$, which discusses the movie "WALL-E" and its pity for not winning the Oscar, and context $c_4$, which covers the history of Pixar studio, both receive CI values close to zero, indicating that their presence does not contribute to answering the question correctly.

### E.3 Case Study 3

Figure 11 presents another illustrative example from the HotpotQA dataset. To answer the question "Which was fought earlier in US's history, the Seven Days Battles or the Battle of Manila?", one must provide accurate information about both historical events. However, the "Battle of Manila" presents an inherent ambiguity due to multiple battles bearing this name throughout history. The question's specification of "in US history" helps narrow the scope, indicating that we should only consider the Battle of Manila that occurred after the United States' establishment. Our analysis reveals that contexts $c_3$ and $c_4$, which discuss the Battle of Manila in Philippine history prior to the US's founding, are potentially misleading and receive negative CI values ($\phi_3(v) = -0.32$ and

**Query**: When did Toyota first come to the United States?
**Golden Answer**: 1957

**Retrieved Contexts**

$c_1$ "… By the end of the decade, Toyota had established a worldwide presence, as the company had exported its one-millionth unit. The first Japanese vehicles to arrive in North America were five Land Cruisers in El Salvador in May 1953. "
$\phi_1(v) = -0.19$

$c_2$ "…Toyota's first export to the United States began with 30 Crown Deluxe's in June 1958 after establishing Toyota Motor Sales USA the previous October, the first directly managed retail dealer (Hollywood Toyota) the previous February, U.S. wholesale and import companies."
$\phi_2(v) = -0.14$

$c_3$ "…However, when Toyota eventually entered the American market in 1957 with the Crown, the name was not well received due to connotations of toys and pets. The name was soon dropped for the American market, but continued in other markets until the mid-1960s…"
$\phi_3(v) = 1.01$

$c_4$ "The history of Toyota started in 1933 with the company being a division of Toyoda Automatic Loom Works devoted to the production of cars under the direction of the founder's son, Kiichiro Toyoda. Kiichiro Toyoda had traveled to Europe and the United States in 1929 to investigate automobile production and had begun researching gasoline-powered engines in 1930. Toyoda Automatic Loom Works was encouraged to develop automobile production by the Japanese government, which needed domestic vehicle production, due to the war with China."
$\phi_4(v) = 0.00$

…

**Prediction w/ full contexts**: May 1953 ❌
**Prediction w/ contexts of positive CI values**: 1957 ✔️

Figure 9: Case study (test case #138 in NQ dataset) with Llama3-8B as LLM generator.

**Query**: Which film won the Oscar for best animated feature in 2007?
**Golden Answer**: Ratatouille

**Retrieved Contexts**

$c_1$ "… Ratatouille won the Best Animated Feature award at the 2008 Golden Globes; it was also nominated for 5 Academy Awards, including Best Animated Feature and Best Original Screenplay. On February 24, 2008, Ratatouille won Bird his second Academy Award for Best Animated Feature Film.. "
$\phi_1(v) = 0.57$

$c_2$ "Golden Globe Award for Best Animated Feature Film The Golden Globe Award for Best Animated Feature Film was awarded for the first time at the 64th Golden Globe Awards in 2007. It was the first time that the Golden Globe Awards had created a separate category for animated films since its establishment. The nominations are announced in January and an awards ceremony is held later in the month. Initially, only three films are nominated for best animated film, in contrast to five nominations for the majority of other awards. The Pixar film Cars was the first recipient of the award"
$\phi_2(v) = -0.14$

$c_3$ "At the 81st Academy Awards, in which WALL-E won the award but was not nominated for Best Picture, despite receiving widespread acclaim from critics and audiences and being generally considered one of the best films of 2008…"
$\phi_3(v) = -0.02$

$c_4$ "…In 2007, Pixar released Meet the Robinsons, which experienced a poor response at the box office despite the lukewarm critical and audience reception. The following film, 2008's Bolt had the best critical reception of any Disney animated feature since Lilo & Stitch, and became a moderate success."
$\phi_4(v) = 0.03$

…

**Prediction w/ full contexts**: Cars ❌
**Prediction w/ contexts of positive CI values**: Ratatouille ✔️

Figure 10: Case study (test case #143 in NQ dataset) with Llama3-8B as LLM generator.

**Query**: Which was fought earlier in US history, the Seven Days Battles or the Battle of Manila?
**Golden Answer**: The Seven Days Battles

**Retrieved Contexts**

$c_1$ "The Seven Days Battles were a series of seven battles over seven days from June 25 to July 1, 1862, near Richmond, Virginia, during the American Civil War. Confederate General Robert E... "
$\phi_1(v) = 0.21$

$c_2$ "The Battle of Manila (Filipino: Labanan sa Maynila; ), the first and largest battle of the Philippine–American War, was fought on February 4–5, 1899, between 19,000 American soldiers and 15,000 Filipino armed militiamen... "
$\phi_2(v) = 0.23$

$c_3$ "The Battle of Manila (Filipino: Labanan sa Maynila) (1365) is an unspecified and disputed battle occurring somewhere in the vicinity of Manila between the forces of the kingdoms in Luzon and the Empire of Majapahit..."
$\phi_3(v) = -0.32$

$c_4$ "...Websites Battles of La Naval de Manila The Battles of La Naval de Manila () were a series of five naval battles fought in the waters of the Philippines in the year 1646, in which the forces of Spain repelled various attempts by forces of the Dutch Republic to invade Manila, during the Eighty Years' War. The Spanish forces, which included many native Kapampángan volunteers, consisted of two, and later, three Manila galleons, a galley and four brigantines..."
$\phi_4(v) = -0.14$

...

**Prediction w/ full contexts**: The Battle of Manila (1365) ✗
**Prediction w/ contexts of positive CI values**: The Seven Days Battles ✓

Figure 11: Case study (test case #157 in HotpotQA dataset) with Llama3-8B as LLM generator.

$\phi_4(v) = -0.14$). These contexts could lead the generator to produce incorrect answers. In contrast, contexts $c_1$ and $c_2$, which contain the correct temporal information about both the Seven Days Battles and the relevant Battle of Manila, are assigned positive CI values ($\phi_1(v) = 0.21$ and $\phi_2(v) = 0.23$). The retention of these contexts enables the generator to produce the correct answer.

