# OpenReview forum: "Influence Guided Context Selection for Effective Retrieval-Augmented Generation"
_NeurIPS.cc/2025/Conference — NeurIPS 2025 poster_

### Official Review · Reviewer_ynDW · 2025-07-01

**Clarity:** 2
**Significance:** 2
**Originality:** 2
**Rating:** 4
**Confidence:** 4

**Summary:**

This paper propose to assess context in RAG via inference-time data valuation, and a contextual influence value(CI) is proposed.  Specifically, a parameterized surrogate model is trained to predict CI values of difference contexts by capturing dependences both  between query-context and inter-contexts, and the proposed method simply retains contexts with positive CI values. Experimental results show that the proposed method can achieve some performance improvement.

**Questions:**

Please see Weaknesses

**Ethical Concerns:**

["NO or VERY MINOR ethics concerns only"]

**Final Justification:**

The rebuttal addressed some of my concerns and I adjust my score accordingly.

**Limitations:**

yes

**Quality:**

2

**Strengths And Weaknesses:**

Strengths:
1. The idea of measuring a context via the performance degradation when removing it is reasonable;
2. CI values can take all query-aware, list-aware, and generator-aware metrics into consideration;
3. Experimental results show some effectiveness of the proposed method.

 Weaknesses:
1. My main concern about this paper is the mismatch between the motivation of CI value( the performance degradation when removing a specific context) and how it is actually calculated(a surrogate model which simply takes several input and output the CI value). It is hard to say such a model can match the motivation of the CI value, and whether can generalize to unseen scenarios. Therefore, I think the authors should explain why such a surrogate model can actually reflect the CI value, and give detailed analysis between golden CI values and calculated value using several examples.
2. The current baselines are mainly vanilla RAG + context scoring, the authors should also compared with advanced RAG systems such as agentic RAG systems, which can also take the quality and the importance of contexts into consideration. To make the experimental results more convincing, the authors should also compare with state-of-the-art solutions, rather than only simple baselines.

---

> ### Author Rebuttal · Authors · 2025-07-29
>
> > **Weakness 1: My main concern about this paper is the mismatch between the motivation of CI value( the performance degradation when removing a specific context) and how it is actually calculated(a surrogate model which simply takes several input and output the CI value). It is hard to say such a model can match the motivation of the CI value, and whether can generalize to unseen scenarios. Therefore, I think the authors should explain why such a surrogate model can actually reflect the CI value, and give detailed analysis between golden CI values and calculated value using several examples.**
>
> **Response**: The motivation of developing CI value as the context quality assessment metric is that it satisfies four key properties: query-awareness (context-query relevance), list-awareness (context-context dependance), generator-awareness (explicit feedback from generator) and ease-of-configuration. These properties are crucial for improving context selection performance, but they cannot be simultaneously satisfied by any existing metrics.
>
> Despite the favorable properties of CI value, it is infeasible to exactly calculate CI value in the inference time, since (1) CI value requires golden answers for computing utilities, which are unavailable during inference; and (2) calculating CI value requires multiple LLM forwards, leading to significant computational overhead. To make it feasible, we propose a parameterized model (i.e., CSM) to approximate CI values.
>
> To match the motivation of CI value, we have carefully designed the architecture and training paradigm of the CSM.  To ensure query-awareness and list-awareness, we developed a hierarchical model structure by capturing both query-context semantics and context list-wise semantics, as illustrated in Figure 2(a). To inject generator-awareness into CSM, we explored two training strategies: (1)  supervised learning using oracle CI values as targets, which provides implicit generator feedback;  and (2) end-to-end training with the generator in the loop, which offers explicit signals about each context’s impact.
>
> To evaluate how these designs help approximating CSM's predictions with golden CI values, we conducted empirical studies in Section 6.2: (1) By comparing the RAG performance of the CSM with other baselines, we show that the CSM performs effectively in context selection, closely approximating the performance of the oracle CI value; (2) The Spearman correlation between CSM's predictions and golden CI values is high; and (3) Ablation studies by corrupting the training strategies confirm the necessity of our designs.
>
>
>
> > **Weakness 2: The current baselines are mainly vanilla RAG + context scoring, the authors should also compared with advanced RAG systems such as agentic RAG systems, which can also take the quality and the importance of contexts into consideration. To make the experimental results more convincing, the authors should also compare with state-of-the-art solutions, rather than only simple baselines.**
>
> **Response**: We would like to clarify that advanced RAG pipelines are also included as baselines in our experiments. Take the baseline Ret-Robust [1] for example, it is implemented as a self-ask RAG pipeline[2] that use an agentic LLM to decompose complex questions into subquestions and perform multi-round of retrieval and generation until generating the final answer. It is one of the state-of-the-art baselines in the latest FlashRAG benchmark [2].
>
> For more comprehensive evaluation, we conduct additional experiments using more advanced RAG pipelines as baselines. We first provide a brief introduction to the new baselines:
>
> - Self-RAG [3] performs on-demand retrieval and learns to reflect on retrieved contexts while critiquing generated answers.
>
> - RQ-RAG [4] enhances the RAG pipeline through explicit rewriting, decomposition, and disambiguation.
>
> These baselines encompass common agentic capabilities including reflection and planning. The following table presents the RAG performance (using Llama3-8B-Instruct as generator) of CSM compared to these baselines.
>
> |              | NQ (EM)  | TriviaQA (EM) | WebQA (EM) | HotpotQA (F1) |
> | ------------ | :------: | :-----------: | :--------: | :-----------: |
> | Self-RAG [3] |   0.36   |     0.38      |    0.22    |     0.30      |
> | RQ-RAG [4]   |   0.34   |     0.55      |  **0.26**  |     0.35      |
> | CSM-st       | **0.42** |   **0.69**    |    0.24    |   **0.47**    |
>
> The results demonstrate that CSM achieves strong performance across most datasets, outperforming state-of-the-art agentic baselines on NQ, TriviaQA, and HotpotQA, while maintaining competitive performance on WebQA. These results highlight CSM's effectiveness in filtering out low-quality contexts in RAG, when compared to sophisticated agentic approaches that employ complex reflection and planning mechanisms.

---

> > ### Author Response · Authors · 2025-08-01
> > **reference**
> >
> > [1] Yoran, Ori, et al. "Making Retrieval-Augmented Language Models Robust to Irrelevant Context." *The Twelfth International Conference on Learning Representations*.
> >
> > [2] Jin, Jiajie, et al. "Flashrag: A modular toolkit for efficient retrieval-augmented generation research." *Companion Proceedings of the ACM on Web Conference 2025*.
> >
> > [3] Asai, Akari, et al. "Self-RAG: Learning to Retrieve, Generate, and Critique through Self-Reflection." *The Twelfth International Conference on Learning Representations*.
> >
> > [4] Chan, Chi-Min, et al. "Rq-rag: Learning to refine queries for retrieval augmented generation." *arXiv preprint arXiv:2404.00610* (2024).

---

> > ### Author Response · Authors · 2025-08-05
> >
> > Dear Reviewer ynDW
> >
> > Thanks for your dedication to reviewing our paper!
> >
> > In our early response, we have included more experiments and clarifications. We would appreciate it if you let us know whether your previous concerns have been adequately addressed. If you have any further questions, please do not hesitate to let us know so that we can respond to them timely.

---

> ### Comment · Area_Chair_yNsP · 2025-08-05
>
> Dear Reviewer ynDW,
>
> Thank you again for your thoughtful review.
>
> As the only reviewer with a negative recommendation, your perspective is especially important. The authors have responded in detail to your concerns.
>
> With the discussion period ending soon, I encourage you to share whether the rebuttal addresses your concerns—or to raise any remaining points for discussion.
>
> Best regards,
> Your AC

---

### Official Review · Reviewer_2KDq · 2025-07-02

**Clarity:** 3
**Significance:** 3
**Originality:** 3
**Rating:** 4
**Confidence:** 4

**Summary:**

This paper proposes a new reranking model for context selection for retrieval augmented generation (RAG). Specifically, an auxiliary model is trained to predict how much removing a context from a list of retrieved documents will affect the final performance, i.e. CI or contextual influence value. Two training strategies are introduced, one based on supervised signals derived from labeled data, and one based on generated data. Specific training tricks are proposed to deal with imbalanced distributions of CI values. Experiments show the effectiveness of the CI value-based context selection performance as of the final answer quality.

**Questions:**

- Due to the nature of CI values, it is okay and reasonable to select context based on whether the CI value is positive. However, for fairness of comparison, it is better to ensure there are no significantly more contexts being selected. Maybe also report the average number of contexts selected based on CI
- In figure 3, it clearly shows the performance first increases and then decreases. I thus wonder whether the best performance always corresponds to the zero points of CI values empirically?
- Iine 258-259, it is mentioned the data splits used by several baselines. However, it is not mentioned whether the proposed method also use the same splits.

**Ethical Concerns:**

["NO or VERY MINOR ethics concerns only"]

**Final Justification:**

Overall, this paper proposes an interesting idea and the writing is clear. My concerns are addressed well and the added information should be updated in the paper draft. I would keep my score of a Borderline accept.

**Limitations:**

yes

**Quality:**

3

**Strengths And Weaknesses:**

### Strengths
- The idea of how each context contributes to the final answer is meaningful for RAG and is studied by previous research in different forms, as also mentioned in the paper.
- Experiments clearly show the induced CI value correlates to the answer quality and thus useful as reranking scores
- The methodology is clearly described with plenty of details.

### Weaknesses
- The way of using CI scores during inference could be better analyzed. see some questions below
- It would be better to include some efficiency comparison on some compared methods to show the applicability of the method.

---

> ### Author Rebuttal · Authors · 2025-07-29
>
> > **Weakness 1 and Question 1: Due to the nature of CI values, it is okay and reasonable to select context based on whether the CI value is positive. However, for fairness of comparison, it is better to ensure there are no significantly more contexts being selected. Maybe also report the average number of contexts selected based on CI**
>
> **Response**: For fair comparison, the number of selected contexts (top-$k$) is kept consistent across all baselines. Since CSM selects all positive contexts, we set top-$k$ as the average number of positive contexts, to ensure that the number of selected contexts is equivalent between CSM and the baselines. The table below reports the average number of positive contexts (denoted as $k_{\rm pos}$) when retrieving 10 contexts:
>
> |               |  NQ  | TriviaQA | WebQA | HotpotQA | 2Wiki | FEVER | TruthfulQA | ASQA |
> | ------------- | :--: | :------: | :---: | :------: | :---: | :---: | :--------: | :--: |
> | $k_{\rm pos}$ | 4.7  |   4.9    |  4.7  |   5.5    |  5.4  |  5.5  |    4.5     | 5.4  |
>
> Based on the above table, we set top-$k=5$ for all baselines across all datasets.
>
> > **Weakness 2: It would be better to include some efficiency comparison on some compared methods to show the applicability of the method.**
>
> **Response**: The efficiency of RAG systems primarily depends on the parameter size of the context selection model when keeping the retriever and generator the same. The table below presents the parameter sizes of context selection models and their corresponding CUDA times (from context selection to generation) for different baselines w.r.t. the number of retrieved contexts $n$. Since CSM selects all positive contexts, for fair comparison, we set the number of preserved contexts (i.e., top-$k$) for the baselines equal to the average number of contexts with positive CI values (i.e., $k_{\rm pos}$). For $n=10$, we set top-$k=k_{\rm pos}=5$, and for $n=50$, we set top-$k=k_{\rm pos}=23$. The experiments utilize E5-base-v2 as the retriever, Llama3-8B-Instruct as the generator, and are conducted on a server equipped with Montage Jintide(R) C6226R CPU, 256GB Memory, and 4 NVIDIA GeForce RTX 4090 GPUs.
>
> |              | Parameter Size | NQ ($n$=10) | NQ ($n$=50) | TriviaQA ($n$=10) | TriviaQA ($n$=50) | HotpotQA ($n$=10) | HotpotQA ($n$=50) |
> | :----------- | -------------- | :---------: | :---------: | :---------------: | :---------------: | :---------------: | :---------------: |
> | Standard RAG | -              |   320 ms    |   811 ms    |      252 ms       |      810 ms       |      261 ms       |      814 ms       |
> | RankGPT      | 8030M          |   874 ms    |   1437 ms   |      779 ms       |      1561 ms      |      741 ms       |      1640 ms      |
> | RECOMP-abs   | 775M           |   299 ms    |   662 ms    |      254 ms       |      721 ms       |      202 ms       |      924 ms       |
> | CSM          | 140M           |   253 ms    |   481 ms    |      192 ms       |      402 ms       |      206 ms       |      423 ms       |
>
> Compared to Standard RAG, CSM can reduce the input length of generator without introducing heavy computation overhead, thereby decreasing the overall computation time. Compared to baseline context selection models, CSM achieves significantly lower end-to-end latency due to its lightweight model architecture. This efficiency advantage becomes more pronounced as the number of retrieved contexts increases from $n=10$ to $n=50$. For instance, on NQ with $n=50$, CSM reduces the latency to 481 ms, while RankGPT requires 1437 ms and RECOMP-abs 662 ms. This demonstrates that CSM scales more efficiently when $n$ grows large.
>
> > **Question 2: In figure 3, it clearly shows the performance first increases and then decreases. I thus wonder whether the best performance always corresponds to the zero points of CI values empirically?**
>
> **Response**: Let $k_{\rm zero}$ be the zero point of CI values, the best performance always corresponds to $k_{\rm zero}$ or $k_{\rm zero}-1$ in our experiments. When plotting Figure 3, we set $k_{\rm zero}$ as the nearest integer, e.g., if the average number of positive contexts is 4.7, we round it to $k_{\rm zero} = 5$. Then, for the context selection experiments by adding top-$k_{\rm zero}$ high-quality contexts, some negative contexts may also be selected, thus affecting the performance.
>
>
> > **Question 3: Iine 258-259, it is mentioned the data splits used by several baselines. However, it is not mentioned whether the proposed method also use the same splits.**
>
> **Response**: Thank you for pointing this out. The data splits are the same for both baselines and the proposed CSM, and we will clarify this in the subsequent version of the paper.

---

> > ### Author Response · Authors · 2025-08-05
> >
> > Dear Reviewer 2KDq
> >
> > Thanks for your dedication to reviewing our paper!
> >
> > In our early response, we have included more experiments and clarifications. We would appreciate it if you let us know whether your previous concerns have been adequately addressed. If you have any further questions, please do not hesitate to let us know so that we can respond to them timely.

---

> > ### Comment · Reviewer_2KDq · 2025-08-05
> >
> > Thanks for the detailed explanation. I do not have further questions. I think my initial score is consistent with my final judgement.

---

### Official Review · Reviewer_77ay · 2025-07-02

**Clarity:** 2
**Significance:** 3
**Originality:** 3
**Rating:** 4
**Confidence:** 4

**Summary:**

The author proposes a method for automatic context document selection in the RAG scenario called CSM. Specifically, the author introduces the use of Contextual Influence (CI) values to measure the usefulness of contexts. Two training methods, Supervised training and end-to-end training, are employed to train the CSM model to predict context CI values. During inference, CSM predicts the CI values for all contexts and selects those with CI values greater than 0. The author provides extensive experimental evidence demonstrating the effectiveness of CSM.

**Questions:**

1. I am concerned whether the oracle CI scores in certain special cases accurately represent the usefulness of the context. For example, when a user queries information about a hot news topic, the retriever may retrieve several similar contexts. These contexts provide the same important information, and removing one does not affect the utility. According to Definition 1, their CI values will be near 0, but these contexts are still quite important. Does this represent a potential flaw?
2. In the experiments, is the CSM model the same for different datasets? How does the CSM model perform on out-of-distribution data? This is an important concern.
3. In the End-to-end Training of CSM section, the authors use a masking method to obtain $H_c$. This could introduce a potential issue: during training, low-scoring context positions in $H_c$ still have placeholders close to zero, but these placeholders do not exist during inference, leading to a gap between training and inference. I understand this is a trade-off between differentiability and efficiency, but I am still curious whether this gap affects performance.

Minor Comments:
1. The average improvement in Table 1 is quite misleading when presented as a percentage. It is recommended to include the absolute value of the increase as well.

**Ethical Concerns:**

["NO or VERY MINOR ethics concerns only"]

**Limitations:**

yes

**Quality:**

3

**Strengths And Weaknesses:**

Strengths:
1. The motivation is clear, and CSM efficiently and effectively avoids the manual selection of top-k documents, which is very attractive.
2. The end-to-end training method for CSM is ingenious and successfully addresses the issue of the context selection process being discrete and non-differentiable.
3. The author provides the complete code.

Weakness:
1. The method section may have some potential flaws
2. The performance on OOD data has not been explored.
Please see the Questions section for details

---

> ### Author Rebuttal · Authors · 2025-07-29
>
> > **Weakness 1: The method section may have some potential flaws.**
> >
> > **Question 1: I am concerned whether the oracle CI scores in certain special cases accurately represent the usefulness of the context. For example, when a user queries information about a hot news topic, the retriever may retrieve several similar contexts. These contexts provide the same important information, and removing one does not affect the utility. According to Definition 1, their CI values will be near 0, but these contexts are still quite important. Does this represent a potential flaw?**
>
> **Response**: We would like to clarify that the definition of CI value is build on the assumption that there are no redundant context in the retrieved context list. This assumption can be easily satisfied in modern vector databases. For example, faiss support vector deduplication during KNN search. To further prevent highly similar contexts from affecting the correctness of CI values, we adopt a simple yet effective mechanism inspired by previous works [1, 2]: we preprocess the retrieved contexts by clustering semantically similar contexts, and assign the same CI value to contexts inside each cluster. In our experiments, this mechanism is rarely triggered, occurring in only about 3% of the data samples.
>
>
>
> > **Weakness 2: The performance on OOD data has not been explored.**
> >
> > **Question 2: In the experiments, is the CSM model the same for different datasets? How does the CSM model perform on out-of-distribution data? This is an important concern.**
>
> **Response**: The CSM model is not shared across different datasets. CSM is trained to fit CI value distributions and approximate CI values under specific inference tasks. Therefore, if a different dataset leads to CI value distribution shift, the CSM model needs to be retrained accordingly. Please note that the compared context selection methods similarly require retraining on each new dataset, as their performance is also sensitive to distributional shifts.
>
> Following your suggestion, we simulate out-of-distribution (OOD) scenarios by training CSM on one dataset (e.g., NQ) and evaluating it on a different dataset (e.g., TriviaQA). In contrast, in-distribution (ID) scenarios involve training and evaluating CSM on the same dataset. The following table shows the results in the form of $a\to b$, where a is the RAG performance under ID setting and b is the RAG performance under OOD setting.
>
> |            | Parameter size | TriviaQA (EM) |  WebQA (EM)   | HotpotQA (F1) |
> | ---------- | :------------: | :-----------: | :-----------: | :-----------: |
> | RECOMP-ex  |      109M      | 0.60$\to$0.55 | 0.18$\to$0.14 | 0.39$\to$0.35 |
> | RECOMP-abs |      775M      | 0.59$\to$0.57 | 0.18$\to$0.15 | 0.40$\to$0.37 |
> | CSM-st     |      140M      | 0.69$\to$0.61 | 0.24$\to$0.19 | 0.47$\to$0.35 |
>
> We can observe a consistent performance drop across all methods under OOD setting, but **CSM still outperforming other baselines**. Meanwhile, the performance drop in HotpotQA is larger than that in TriviaQA and WebQA. This is because HotpotQA is a multi-hop QA task that quite different from NQ, while TriviaQA and WebQA are both open-domain QA tasks more similar to NQ.
>
> To improve the generalization ability of CSM as well as other context selection methods,  existing techniques for handling out-of-distribution data may be leveraged. For example, training a diverse mixture of tasks may help CSM capture transferable patterns. Additionally, methods like continual or meta-learning can be explored to adapt CSM to new distributions with minimal retraining.
>
>
>
> > **Question 3: In the End-to-end Training of CSM section, the authors use a masking method to obtain $H_c$. This could introduce a potential issue: during training, low-scoring context positions in still have placeholders close to zero, but these placeholders do not exist during inference, leading to a gap between training and inference. I understand this is a trade-off between differentiability and efficiency, but I am still curious whether this gap affects performance.**
>
> **Response**: This gap leads to little affect on the overall performance, since the generator LLM is generalizable to such gap. To evaluate this, we conducted the following experiment: for a data sample $x = (q, C, M)$, where $q$ is the question, $C$ is the retrieved context list, and $M$ is a near-binary mask produced by the CSM indicating which contexts to preserve, we compare the difference of LLM's output $f(x)$ when using soft-selection and hard-selection. For **soft-selection**, we apply the mechanism described in Section 4.3 and obtain the generated answer $\hat{y}_1$. For **hard-selection**, we first perform context selection based on $M$, use the selected contexts as input to the generator, and obtain the generated answer $\hat{y}_2$. We then compute the F1 score between $\hat{y}_1$ and $\hat{y}_2$. A higher F1 score indicates the output of soft selection is similar to that of hard selection, meaning that the generator's behavior is consistent during training and inference. Using test samples from NQ and HotpotQA, we calculate the average F1 score, as shown in the following table:
>
> |            | Llama3-8b-instruct | Qwen2.5-7b-instruct |
> | ---------- | :----------------: | :-----------------: |
> | Average F1 |        0.95        |        0.93         |
>
> Results show that soft-selection and hard-selection leads to highly similar LLM outputs, thus the gap between training and inference has little effect on performance.
>
>
>
> > **Minor Comments: The average improvement in Table 1 is quite misleading when presented as a percentage. It is recommended to include the absolute value of the increase as well.**
>
> **Response**: We report performance improvements in percentage terms to intuitively show the effectiveness of CSM. The percentage is calculated by comparing the average improvement of CSM (both CSM-st and CSM-e2e) ove all baselines (except Vanilla LLM). Absolute improvements will also be included in the subsequent version of the paper.
>
>
>
> [1] He, Yexiao, et al. "Shed: Shapley-based automated dataset refinement for instruction fine-tuning." *Advances in Neural Information Processing Systems* 37 (2024): 99382-99403.
>
> [2] Tarun, Ayush, et al. "EcoVal: An Efficient Data Valuation Framework for Machine Learning." *Proceedings of the 30th ACM SIGKDD Conference on Knowledge Discovery and Data Mining*. 2024.

---

> > ### Comment · Reviewer_77ay · 2025-08-05
> >
> > Thank you for your response. It has addressed my concerns regarding Questions 1 and 3. My current concern is the generalization ability of the CSM model in out-of-distribution (OOD) scenarios, which is Question 2. This concern was also raised by reviewer RASc. The rebuttal from the authors highlights that the CSM model has relatively poor generalization in OOD scenarios. In real-world industrial production environments, user queries and external knowledge have a very wide distribution, so I believe this question should be taken seriously.
> > Overall, this work still makes a solid contribution, and I remain positive about this submission.

---

> > > ### Author Response · Authors · 2025-08-06
> > >
> > > Thank you very much for your constructive feedback.
> > >
> > > We would like to note that in practice, retraining is a standard procedure when deploying models to new tasks or adapting to data distribution shifts. For instance, recommendation systems frequently update their models as user preferences evolve. Similarly, language models also require fine-tuning for new use cases. Once trained for a specific task, CSM provides sustained benefits throughout the model's deployment lifecycle without requiring additional retraining, unless the underlying task or data distribution changes. Notably, in such cases, any production model would necessitate updates.
> > >
> > > When CSM is trained on sufficiently comprehensive and representative data covering the expected distributions of queries and knowledge, retraining becomes unnecessary. Out-of-distribution (OOD) scenarios typically arise from a misalignment between training data and the deployment environment. In industrial settings, this challenge can often be addressed by carefully curating training datasets that encompass the anticipated range of real-world use cases.
> > >
> > > We agree that further improving CSM's robustness to truly unseen distributions remains valuable future work, and we appreciate you highlighting this important consideration.

---

### Official Review · Reviewer_RASc · 2025-07-03

**Clarity:** 4
**Significance:** 3
**Originality:** 4
**Rating:** 4
**Confidence:** 4

**Summary:**

This paper studies context selection in Retrieval-Augmented Generation, where Large Language Models use retrieved knowledge to reduce hallucination. The authors identify key limitations in existing context selection methods, which typically rely on query, list, or generator-aware quality metrics, often failing to comprehensively balance these factors and requiring dataset-specific hyperparameter tuning. To address these issues, the paper introduces the Contextual Influence (CI) value, which quantifies each context’s contribution by measuring the drop in performance when it is removed. They further propose a Context Selection Model (CSM) that predicts CI values at inference time without needing access to labels, using hierarchical modeling and either supervised or end-to-end training with generator feedback. Experiments on eight NLP tasks and two LLM backbones show that this approach outperforms a wide range of strong baselines, leading to improved generation quality and robustness.

**Questions:**

1.How would the CI value be adapted if the generation utility (e.g., $v(f(q⊕C))$) is not well-defined by EM/F1 (e.g., creative writing, summarization)?

2.In scenarios where useful information is distributed across several weakly helpful contexts (none with individually positive CI), could the current approach lead to information loss? Are there strategies to mitigate filtering out such complementary but individually weak contexts?

3.How well does the CI surrogate model generalize to new retrieval domains? Would it require retraining for each new task, or is it robust enough to handle domain shifts without additional tuning?

**Ethical Concerns:**

["NO or VERY MINOR ethics concerns only"]

**Final Justification:**

The authors have made the following responses or adjustments:
1.Supplemented the experiments with inference time cost analysis;
2.Responded to the concern regarding the diversity of application scenarios.
However, the CSM method still exhibits notable limitations, such as the need for retraining on new tasks. Therefore, I have decided to keep my current score unchanged.

**Limitations:**

yes

**Quality:**

3

**Strengths And Weaknesses:**

> Strengths：

1. The CI value offers a holistic context evaluation metric that explicitly combines query relevance, listwise context uniqueness, and generator feedback—overcoming key limitations of previous metrics.

2. The proposed CSM model structurally encodes both local (query-context) and global (context-context) dependencies, a sound design that reflects the nature of the selection problem.

3. The paper presents consistent improvements over a broad range of baselines—including query-aware, list-aware, and generator-aware methods, as well as robust LLM tuning—across various QA tasks. The evaluation is comprehensive, combining diverse benchmarks with both quantitative metrics and qualitative case studies to support the claims.

> Weaknesses：

1.There is little quantitative comparison of inference-time latency or computational resources used by CSM versus standard RAG and other selection approaches, especially as the number of retrieved contexts grows large.

2.While the method elegantly sidesteps the cost of explicit influence computation, details on CSM’s potential to misestimate context dependencies in more adversarial or correlated inference setups are limited.

---

> ### Author Rebuttal · Authors · 2025-07-29
>
> > **Weakness 1: There is little quantitative comparison of inference-time latency or computational resources used by CSM versus standard RAG and other selection approaches, especially as the number of retrieved contexts grows large.**
>
> **Response:** The efficiency of RAG systems primarily depends on the parameter size of the context selection model when keeping the retriever and generator the same. The table below presents the parameter sizes of context selection models and their corresponding CUDA times (from context selection to generation) for different baselines w.r.t. the number of retrieved contexts $n$. Since CSM selects all positive contexts, for fair comparison, we set the number of preserved contexts (i.e., top-$k$) for the baselines equal to the average number of contexts with positive CI values (i.e., $k_{\rm pos}$). For $n=10$, we set top-$k=k_{\rm pos}=5$, and for $n=50$, we set top-$k=k_{\rm pos}=23$. The experiments utilize E5-base-v2 as the retriever, Llama3-8B-Instruct as the generator, and are conducted on a server equipped with Montage Jintide(R) C6226R CPU, 256GB Memory, and 4 NVIDIA GeForce RTX 4090 GPUs.
>
> |              | Parameter Size | NQ ($n$=10) | NQ ($n$=50) | TriviaQA ($n$=10) | TriviaQA ($n$=50) | HotpotQA ($n$=10) | HotpotQA ($n$=50) |
> | :----------- | -------------- | :---------: | :---------: | :---------------: | :---------------: | :---------------: | :---------------: |
> | Standard RAG | -              |   320 ms    |   811 ms    |      252 ms       |      810 ms       |      261 ms       |      814 ms       |
> | RankGPT      | 8030M          |   874 ms    |   1437 ms   |      779 ms       |      1561 ms      |      741 ms       |      1640 ms      |
> | RECOMP-abs   | 775M           |   299 ms    |   662 ms    |      254 ms       |      721 ms       |      202 ms       |      924 ms       |
> | CSM          | 140M           |   253 ms    |   481 ms    |      192 ms       |      402 ms       |      206 ms       |      423 ms       |
>
> Compared to baseline context selection models, CSM achieves significantly lower latency due to its lightweight model architecture. This efficiency advantage becomes more pronounced as the number of retrieved contexts increases from $n=10$ to $n=50$. For instance, on NQ with $n=50$, CSM reduces the latency to 481 ms, while RankGPT requires 1437 ms and RECOMP-abs requires 662 ms. This demonstrates that CSM scales more efficiently when $n$ grows large.
>
> > **Weakness 2: While the method elegantly sidesteps the cost of explicit influence computation, details on CSM’s potential to misestimate context dependencies in more adversarial or correlated inference setups are limited.**
>
> **Response:** We appreciate the reviewer's concern regarding CSM’s behavior under adversarial or correlated inference setups. CSM is trained to fit the context influence (CI) value distributions without making assumptions about the quality of retrieved contexts. When trained on CI values computed from adversarial or correlated inference data, CSM naturally learns to estimate context utility in such challenging scenarios. Our experiments include training samples from both types of setups. Adversarial conditions, where noisy or distracting contexts are common, are frequently observed in open-domain QA datasets such as NQ, TriviaQA, and WebQA [1]. Correlated setups, where evidence is distributed across multiple contexts, are typical in multi-hop QA tasks like HotpotQA. Moreover, the hierarchical architecture of CSM is specifically designed to address these challenges by jointly modeling query-context interactions, which help identify and down-weight noisy contexts, and context-context interactions, which capture dependencies among correlated contexts. These design choices enable CSM to select high-utility contexts even in the presence of noise and complex inter-context relationships.
>
>
>
> > **Questions 1: How would the CI value be adapted if the generation utility (e.g.,$v(f(q⊕C))$) is not well-defined by EM/F1 (e.g., creative writing, summarization)?**
>
> **Response**: For any context set $S$, the utility function $v(A)=v(f(q\oplus S))$ measures the quality of the generated answer $A$ when taking $q\oplus S$ as input. The CI value of a context quantifies the decrease in answer quality when that context is removed from the context list. Therefore, as long as answer quality can be effectively evaluated, the CI value can be readily adapted.
>
> When golden answers are available, real-valued scores (e.g., EM/F1/perplexity) serve as well-qualified metrics for measuring the quality of generated answers. For tasks without golden answers (e.g., creative writing and summarization), LLM-as-a-Judge metrics [2] provide a feasible alternative. Specifically, let $f_e$ denote the LLM-based evaluator, based on predefined quality dimensions (e.g., clarity, coverage, etc. [3]), it can either assign quality scores to any given answer or directly compare two candidate answers. The adaptation of CI value $\phi_i$ for context $c_i$ proceeds as follows:
>
> - If the LLM-based evaluator assigns quality scores to answers, i.e., $f_e(A)\in \mathbb{R}$, then $\phi_i=f_e(A_C)-f_e(A_{C\backslash\{i\}})$, where $A_C$ and $A_{C\backslash\{i\}}$ represent the generated answers with and without context $c_i$, respectively.
>
> - If the LLM-based evaluator directly compares two candidate answers, i.e., $f_e(A_1,A_2)\in \{0,1\}$, where $f_e(A_1,A_2)=0$ indicates that $A_1$ is superior to $A_2$, then $\phi_i=f_e(A_C,A_{C\backslash\{i\}})$, where $A_C$ and $A_{C\backslash\{i\}}$ represent the generated answers with and without context $c_i$, respectively.
>
>
>
> > **Question 2: In scenarios where useful information is distributed across several weakly helpful contexts (none with individually positive CI), could the current approach lead to information loss? Are there strategies to mitigate filtering out such complementary but individually weak contexts?**
>
> **Response**: We would like to clarify that weakly helpful contexts may have relatively small but still positive CI values. When useful information is distributed across such contexts, removing any one of them can lead to a decrease in overall utility. We can simply select all contexts with positive CI values to preserve these weakly helpful contexts. Weakly helpful contexts are commonly observed in multi-hop QA tasks like HotpotQA, where answering a question requires information from multiple contexts. To effectively handle these cases, CSM is designed to capture context-context interactions, enabling it to better model the CI value distributions of weakly helpful contexts, thus leading to remarkable performance improvement in multi-hop QA tasks.
>
>
>
> > **Question 3: How well does the CI surrogate model generalize to new retrieval domains? Would it require retraining for each new task, or is it robust enough to handle domain shifts without additional tuning?**
>
> **Response**: CSM currently requires retraining for new, unseen tasks. CSM is trained to fit CI value distributions and approximate CI value under specific tasks, and additional tuning is required when domain shift leads to CI value distribution shift. To improve generalizability across domains, a possible way is to train a larger model with better generalization ability, e.g., using mixture of experts architecture. However, this approach typically incurs substantially higher training costs, and it may still be vulnerable to domain shifts. For these reasons, we prioritize retrain a lightweight CSM model to achieve the goal of fast adaptation.
>
> [1] Fang, Feiteng, et al. "Enhancing Noise Robustness of Retrieval-Augmented Language Models with Adaptive Adversarial Training." *Proceedings of the 62nd Annual Meeting of the Association for Computational Linguistics (Volume 1: Long Papers)*. 2024.
>
> [2] Gu, Jiawei, et al. "A survey on llm-as-a-judge." *arXiv preprint arXiv:2411.15594* (2024).
>
> [3] Nguyen, Huyen, et al. "A comparative study of quality evaluation methods for text summarization." *arXiv preprint arXiv:2407.00747* (2024).

---

> > ### Comment · Reviewer_RASc · 2025-08-05
> >
> > Thank you for the response, which addressed my concern regarding the time cost. However, I believe that the need for retraining on new tasks imposes a significant limitation on the applicability of the CSM method. Overall, I decide to keep my current score unchanged.

---

> ### Author Response · Authors · 2025-08-06
>
> Thank you for your timely and thoughtful feedback.
> We would like to note that in practice, retraining is standard when deploying models to new tasks or when data distribution shifts. For instance, recommendation systems continuously update their models as user preferences evolve. Similarly, deployed language models are regularly fine-tuned for new use cases. Once CSM is trained for a specific task, it continues to provide benefits throughout the model's deployment lifecycle without requiring additional retraining unless the underlying task or data distribution changes, at which point any production model would need updating regardless.
> Thanks again for highlighting this point, we believe that developing a universal context selection model that is generalizable to new tasks is also a promising direction.

---

### Note · Authors · 2025-08-13

We thank the reviewers and AC for their great efforts in evaluating our paper. We summarize the positive points and our responses to the main concerns as follows:

## 1. Positive Points

We sincerely thank the reviewers for recognizing our key contributions:

- Contextual Influence (CI) value is a novel metric for assessing RAG context quality, satisfying crucial principles of query-awareness, list-awareness, generator-awareness, and ease-of-configuration (`RASc`, `77ay`, `2KDq`, and `ynDW`).
- CI Surrogate Model (CSM) offers an effective approximation of CI value when its exact computation is infeasible at inference time (`RASc` and `77ay`).
- CI value is proven highly effective for context ranking (`2KDq`), and CSM consistently outperforms SOTA baselines across diverse NL tasks (`RASc` and `77ay`).

## 2. Concerns

a. Generalization of CSM (`RASc` and `77ay`): We acknowledge that CSM's lightweight design requires retraining for new tasks, which is a standard solution in industrial applications (e.g., recommendation systems) to adapt to data shifts.

To further validate CSM's generalization ability, we present a new experiment. We developed CSM-st-unif, a unified but larger CSM trained on a composite dataset (NQ, TriviaQA, HotpotQA). CSM-st-unif is trained once and directly evaluated on all tasks. We compare it with CSM-st, which retrains a lightweight model for each task.
||NQ(EM)|TriviaQA(EM)|WebQA(EM)|HotpotQA(F1)|2Wiki(F1)|FEVER|
|-----------|:----:|:----------:|:-------:|:----------:|:-------:|:---:|
|CSM-st|0.42|0.69|0.24|0.47|0.26|0.91|
|CSM-st-unif|0.41|0.68|0.21|0.45|0.25|0.91|

Results show CSM-st-unif maintains competitive performance with only a marginal drop even on unseen tasks (WebQA, 2Wiki, FEVER), demonstrating CSM's generalization potential when trained on a diverse dataset.

b. Inference Efficiency of CSM (`RASc` and `2KDq`): CSM's lightweight design makes it faster than large context selection models. This advantage is more pronounced when processing a large number of contexts.

c. Comparison with Agentic RAG (`ynDW`): We have added experiments against agentic RAG like Self-RAG and RQ-RAG, demonstrating that CSM remains superior. Though we did not receive direct feedback from Reviewer `ynDW` during discussion, we hope that our results address the concerns.

d. Setup Clarifications: CI value assumes no redundant contexts (`77ay`) and we set top-k=5 for baselines to match the average number of positive CI contexts for fairness (`2KDq`).

---

### Decision · Program_Chairs · 2025-09-17

**Decision:**

Accept (poster)

**Comment:**

This paper introduces Contextual Influence (CI) value and a Context Selection Model (CSM) for RAG, integrating query-, list-, and generator-awareness into context selection. Reviewers appreciated the clear motivation, lightweight design, and consistent gains across benchmarks, while noting limitations in generalization and the need for retraining on new tasks. The authors strengthened the work in rebuttal with efficiency analyses and comparisons against advanced agentic RAG baselines, which should be included in the final version.